# Celo2: Towards Learned Optimization Free Lunch

**Abhinav Moudgil**[1,2,*]    **Boris Knyazev**[1,3,4]    **Eugene Belilovsky**[1,2]
[1]Mila    [2]Concordia University    [3]Samsung AI Lab, Montreal    [4]Université de Montréal

## Abstract

Learned optimizers are powerful alternatives to hand-designed update rules like Adam, yet they have seen limited practical adoption since they often fail to meta-generalize beyond their training distribution and incur high meta-training cost. For instance, prior work, VeLO, scaled meta-training to 4,000 TPU months ($\sim 10\times$ GPT-3 compute) to meta-train a general-purpose optimizer but it failed to generalize beyond 600M parameters tasks. In this work, we present a surprising finding: by crafting a simple normalized optimizer architecture and augmenting meta-training, it becomes feasible to meta-train a performant general-purpose learned update rule on a tiny fraction of VeLO compute, 4.5 GPU hours to be precise. Our learned update rule scales stably to a billion-scale pretraining task (GPT-3 XL 1.3B) which is six orders of magnitude larger than its meta-training distribution. Furthermore, it shows strong performance across diverse out-of-distribution tasks and is compatible with modern optimization harness that includes orthogonalization, distinct update rules for input-output and hidden weights, and decoupled weight decay. In all, this work paves the way for practically applicable *learnable* optimization algorithms, unlocking exploration of richer meta-training and data curation recipes to further improve performance.

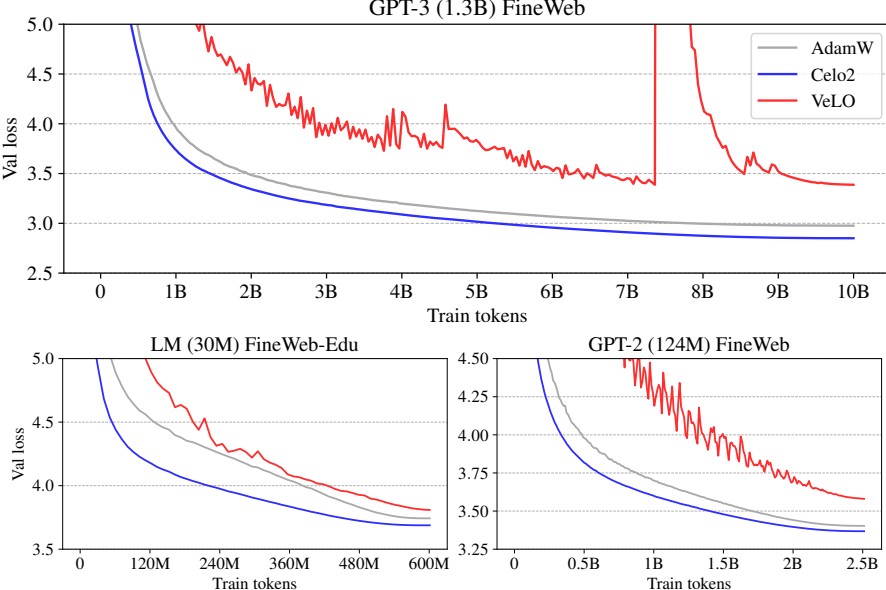

Figure 1: **Celo2, our learned update rule scales stably to large-scale pretraining tasks.** Our learned update is meta-trained on very limited compute budget (4.5 GPU hours) but generalizes to $1{,}000{,}000\times$ larger language modeling task (GPT-3) and outperforms strong well-tuned baselines such as AdamW (Loshchilov, 2017; Kingma & Ba, 2015) and VeLO (Metz et al., 2022b). In contrast, VeLO (Metz et al., 2022b), meta-trained with exorbitant compute (4000 TPU months), fails to generalize. All the language modeling tasks have a modern architecture that includes RMS normalization, rotary positional embeddings (RoPE), QK-norm, and GELU (Marek et al., 2025).

---

*Correspondence to: Abhinav Moudgil <abhinavmoudgil95@gmail.com>

# 1 INTRODUCTION

Pre-training of foundation models has begun to heavily dominate workloads in the past years, producing progressively more powerful models at the cost of great computational resources. Existing optimization strategies used at scale often rely on the now classic Adam optimizer (Kingma & Ba, 2015). As a result, progress made in the direction of improving optimizers for these large-scale tasks thus shows great promise in reducing the computational burden of pre-training. One direction that has yielded strong initial results is learned optimization (Metz et al., 2019b): leveraging learning to improve the optimization process itself. The field of learned optimization has progressively seen more practical results since being introduced in (Andrychowicz et al., 2016). However, progress in this area has not been as fast-paced as that in pre-training, primarily due to the inability of learned optimizers to generalize to data that lies outside their training distributions.

The most powerful learned optimizer to date is the Versatile Learned Optimizer (VeLO) (Metz et al., 2022b), which uses a hierarchical architecture for the network representing the optimizer and then performs meta-learning at an immense scale to achieve generalization across tasks. One key observation regarding VeLO is its high computational requirement: VeLO meta-training used 4000 TPU months' worth of compute to meta-learn the optimizer for achieving high task coverage. It could be argued that this would be partially or fully offset by the fact that VeLO outperformed Adam across a wide variety of tasks, becoming a prototype of the vast potential of learned optimization. However, while the performance gains over Adam were significant, they did not scale to the training of large foundation models like LLMs with billions of parameters. We believe that this result presents a great opportunity for the learned optimization community to tackle the problem of generalization at scale. More recent work (Moudgil et al., 2025; Thérien et al., 2024; Knyazev et al., 2025) in this direction, notably Celo (Moudgil et al., 2025), attempts to address this through architectural modifications and training recipes aimed at improving meta-generalization. While successful in significantly reducing meta-training compute, these approaches show some performance degradation compared to VeLO, which remains the state-of-the-art learned optimizer at scale.

In this work we propose Celo2, a simple recipe for meta-training stable learned optimizers that significantly improves meta-generalization, and is able to outperform VeLO on large-scale tasks while meta-learning on orders of magnitude less. Our proposed recipe aims to tackle the three key challenges: generalization, scalability, and compute-efficiency. We would like to emphasize that our primary objective while developing this simple approach for learned optimization was *stability* (§A): we hope that this work eventually evolves to the level that it can be used directly in the training of foundation models. Finally, we highlight a key finding: we do not explicitly scale up meta-training for performance in order to keep meta-training compute-efficiency, however, we obtain it as a by-product of following the Celo2 recipe, essentially offering a "free lunch" from limited compute.

Our key contributions are as follows:

- We present Celo2, a stable learned optimizer that achieves strong meta-generalization to billion-scale tasks (GPT-3 XL 1.3B), outperforming prior state-of-the-art learned optimizer VeLO (Metz et al., 2022b). Code is available at: `https://github.com/amoudgl/celo2`

- Our proposed meta-training recipe (§3) is compute-efficient and we demonstrate that even with a few GPU hours of meta-training, it demonstrates strong performance on out-of-distribution standard machine learning tasks in language modeling (GPT-2, GPT-3, LM-30M), vision (ViT ImageNet), and reinforcement learning (Atari PPO) domains.

- We conduct systematic ablations keeping meta-generalization as the key focus and also propose Celo2-base, a computationally lighter variant of Celo2 that trades some performance for improved efficiency while maintaining core parts of our generalization recipe.

# 2 RELATED WORK

**Hand-designed optimizers** have long been the cornerstone of neural network training, with methods like SGD with momentum (Sutskever et al., 2013) and Adam (Kingma & Ba, 2015) providing robust baselines through adaptive learning rates and momentum terms. To further enhance convergence and stability, second-order optimizers such as Shampoo (Gupta et al., 2018) introduce preconditioning by maintaining running statistics of gradients in each tensor mode, effectively rescaling

updates to account for curvature in high-dimensional spaces. Similarly, SOAP (Vyas et al., 2024) explores optimization in modular norms, allowing for flexible preconditioning that adapts to the geometry of parameter spaces. Recent advancements have incorporated orthogonalization techniques to mitigate issues like ill-conditioned gradients; for instance, Muon (Jordan et al., 2024) applies a Newton-Schulz iteration to approximately orthogonalize momentum-based updates, projecting them onto the nearest semi-orthogonal matrix to amplify rare directions and improve sample efficiency, as demonstrated in speed-running benchmarks like NanoGPT and CIFAR-10. Complementary work by Tuddenham et al. (2022) proposes gradient orthogonalization via SVD, showing speedups in neural network optimization for SGD with momentum. These approaches highlight the value of preconditioning and orthogonalization in hand-crafted optimizers, but they often require manual tuning and lack the adaptability of learned methods.

**Learned optimizers (LOs)** aim to meta-learn update rules from data, potentially surpassing hand-designed counterparts by discovering task-agnostic strategies. Early efforts, such as those by Andrychowicz et al. (2016), framed optimization as a recurrent process, but scaling remained challenging. State-of-the-art LOs like VeLO (Metz et al., 2022b) represent a breakthrough, training a large hierarchical LO on thousands of diverse tasks to achieve broad generalization, though at immense computational cost (4000 TPU-months). Other works have focused on efficiency: Metz et al. (2022a) analyze trade-offs in LO design, proposing a simple MLP-based optimizer augmented with strong features like Adafac (Adafac MLP) balancing memory and compute while maintaining competitive performance. More recently, $\mu$LOs (Thérien et al., 2024) introduce micro-scale learned optimizers tailored for vision and language models, emphasizing compute-efficient meta-training. These works underscore the potential of LOs but reveal persistent limitations in meta-generalization i.e. the ability to apply learned rules to unseen tasks or longer optimization horizons (unrolls). For example, many LOs struggle with extrapolation beyond meta-training distributions, leading to instability on out-of-distribution problems or when scaling to models with $\geqslant$1B parameters. Celo (Moudgil et al., 2025) addresses some of these by optimizing LO architectures and meta-training protocols for better generalization with minimal compute (24 GPU-hours), outperforming tuned baselines on diverse tasks, yet it stops short of integrating advanced normalization or orthogonalization for further stability. Building on these foundations, our work integrates insights from both hand-designed and learned optimizers. Unlike prior LOs that overlook robust normalization for optimization stability, we introduce a specialized normalization scheme that enhances generalization across tasks and extends unroll lengths. Furthermore, by incorporating the Newton-Schulz orthogonalization from Muon into a learned framework inspired by Celo, we achieve superior meta-generalization and enable stable and performant training of larger models, bridging the gap between efficient meta-training and practical scalability.

## 3  APPROACH

We introduce a new compute-efficient learned optimizer, Celo2 (Alg. 1), that is just meta-trained for 4.5 GPU hours and generalizes to practical large-scale tasks such as GPT-3, ImageNet-ViT, and Reinforcement Learning. The key ingredients of our approach are discussed below:

**Tunable step-size.** Unlike prior work in learned optimization (Metz et al., 2022a;b; Moudgil et al., 2025), our approach *solely focuses* on learning the update rule, thus decoupling step size tuning from update rule learning. This does result in an extra tunable knob in the form of step size. However, we find that with this decoupling, the learned update rule meta-trained on small-scale tasks generalizes surprisingly well to large-scale tasks. This provides a scalable approach to improve the optimization update rule used to train deep networks. This effect of decoupling step size tuning from learning the update rule is also noted by prior work such as Celo (Moudgil et al., 2025). Since Celo learns the scheduler as well as the update rule, it fails to generalize to large-scale tasks. We keep the step size tunable as per the user and leave the learning of a general scheduler for all tasks to future work.

**Simple design that scales.** Building on prior work (Metz et al., 2022a;b; Moudgil et al., 2025), we just learn a small MLP as a drop-in replacement for Adam in large-scale distributed and sharded training setups (§B). We ablate its design in Section 5 to improve meta-generalization and performance, and provide full details on its inputs, outputs, and architecture in Appendix D. The MLP can also be meta-trained on specific tasks; notably, we find that meta-training on simple $8\times8$ image-

---

**Algorithm 1** Celo2

---

**Input:** $\quad\boldsymbol{\theta}_t \qquad$ optimizee parameters
$\qquad\quad\nabla\boldsymbol{\theta}_t \qquad$ gradients
$\qquad\quad\eta_t \qquad$ learning rate
$\qquad\quad\boldsymbol{s}_t \qquad$ optimizer state
$\qquad\quad\alpha \qquad$ (optional) weight decay
**Require:** learned update rule $f_{\text{mlp}}$
$\qquad\qquad$ accumulators update function $f_{\text{acc}}$

1: $\boldsymbol{s}_{t+1} \leftarrow f_{\text{acc}}(\boldsymbol{s}_t, \nabla\boldsymbol{\theta}_t, L_t, t)$ $\qquad\qquad\qquad\qquad\qquad$ $\triangleright$ update accumulators in state
2: initialize next params $\boldsymbol{\theta}_{t+1} \leftarrow ()$
3: **for** each tensor with params $\boldsymbol{p}_t \in \boldsymbol{\theta}_t$ in parallel **do** $\qquad\qquad\qquad$ $\triangleright$ parallel scan over tensors
4: $\quad$ prepare per-param features $\boldsymbol{F}$ for $\boldsymbol{p}_t$ using updated state $\boldsymbol{s}_{t+1}$
5: $\quad\Delta\boldsymbol{p}_t \leftarrow f_{\text{mlp}}(\boldsymbol{F})$ $\qquad\qquad\qquad\qquad\qquad\qquad$ $\triangleright$ learned update rule forward pass
6: $\quad\Delta\boldsymbol{p}_t \leftarrow \text{NewtonSchulz5}(\Delta\boldsymbol{p}_t)$ $\qquad\qquad\qquad\qquad\qquad$ $\triangleright$ Orthogonalization
7: $\quad\Delta\boldsymbol{p}_t \leftarrow \boldsymbol{\Delta}p_t \,/\, \|\Delta\boldsymbol{p}_t\|_{\text{rms}}$ $\qquad\qquad\qquad\qquad\qquad$ $\triangleright$ RMS normalization
8: $\quad\boldsymbol{p}_{t+1} \leftarrow \boldsymbol{p}_t - \eta_t\Delta\boldsymbol{p}_t - \eta_t\alpha\boldsymbol{p}_t$ $\qquad\qquad\qquad\qquad\qquad$ $\triangleright$ update params
9: $\quad\boldsymbol{\theta}_{t+1} \leftarrow \boldsymbol{\theta}_{t+1} \cup \boldsymbol{p}_{t+1}$ $\qquad\qquad\qquad\qquad\qquad\qquad$ $\triangleright$ gather params
$\quad$**return** $\boldsymbol{\theta}_{t+1}, \boldsymbol{s}_{t+1}$

---

classification tasks (§4) suffices to learn a descent rule that generalizes. This provides a scalable path for advancing optimization algorithms.

**Normalized learned update.** Prior work in learned optimizers (Metz et al., 2022a; Harrison et al., 2022), which learn a per-parameter update rule like ours, normalizes inputs $\boldsymbol{F}$ (Alg. 1) to the MLP by RMS of each feature across the parameter tensors. However, all of them directly use raw output from the MLP as the step update. We find that simply normalizing the *outputs* of the MLP update, thus resulting in fully-normalized learned update rule, generalizes well to large-scale tasks. This not only forces the MLP during meta-training to learn a task-invariant update rule (instead of raw MLP outputs) but also results in a similar optimization dynamics as AdamW as demonstrated in Figure 2. The challenges of learning such MLP update rule that operates directly on raw gradients have been discussed in prior work (Almeida et al., 2021; Metz et al., 2019a). Several concurrent works (Liu et al., 2025; Si et al., 2025) also suggest the benefits of using RMS scaling of the step in order to match the learning rate of AdamW which is used to optimize 1D parameters. We extensively ablate over different normalization choices in Section A (appendix) including RMS norm clipping used in AdaFactor (Shazeer & Stern, 2018), RMS accumulation, etc. Empirically, we find that simply normalizing by RMS of the current step works the best. Note that this RMS normalization plays a key role in Celo2-base that uses the learned MLP update for all params; in Celo2, we orthogonalize the MLP update (Step 6, Alg. 1), making it effectively shape-dependent (Liu et al., 2025).

**Compatible with modern optimization harness.** Celo2 is highly compatible with modern techniques such as orthogonalization (Jordan et al., 2024; Gupta et al., 2018), distinct update rules for 1-D and 2-D params (Jordan et al., 2024), and decoupled weight decay (Loshchilov, 2017), as demonstrated by our experiments we describe in the next section. Since Celo2 is a learned update rule, this provides a scalable path to improving optimization algorithms for deep neural networks by simply leveraging the data and compute while building upon and extending recent advances in optimization area.

## 4 GENERALIZATION AT SCALE

**Meta-training.** Following prior work (Moudgil et al., 2025), we meta-train our update rule on 4 image MLP classification tasks consisting of a mixture of datasets such as MNIST (LeCun & Cortes, 1998), Fashion-MNIST (Xiao et al., 2017), CIFAR-10 (Krizhevsky et al., 2009) and SVHN (Netzer et al., 2011). The images are resized to $8\times8$ as in the prior work for faster meta-training, thus yielding a high research throughput. We use Persistent Evolutionary Strategies (PES) (Vicol et al., 2021) to meta-train our optimizer with unroll length logarithmically sampled between 100 and 2000 steps. PES yields unbiased gradients with truncated training. Truncated training which updates the learned optimizer every $K$ steps (we use K=50 in our experiments) during unroll in inner loop allows

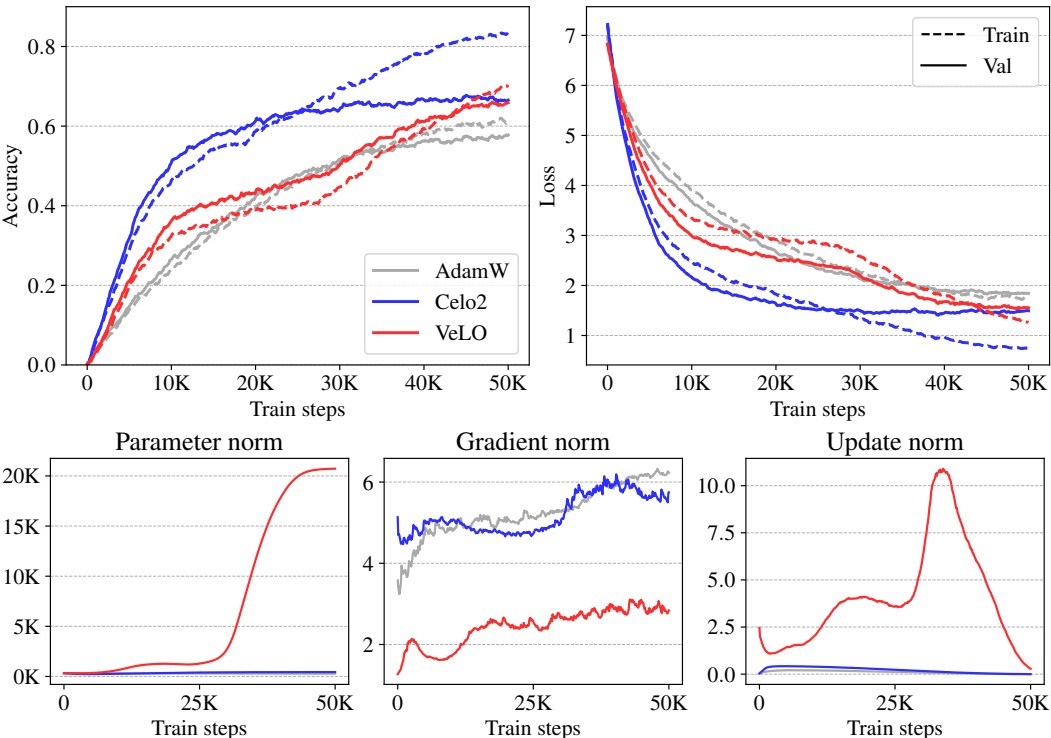

Figure 2: **ImageNet classification with ViT**. We test our learned update rule, Celo2, on the ImageNet classification task with batch size 512 and 50K steps, which is 25× longer than its meta-training unroll length and 30,000× larger than the tasks seen during training. Since VeLO is trained with final loss as the meta-objective, it shows non-trivial dynamics during training in order to achieve low final loss (see norm plots). Celo2 achieves VeLO's final loss within ∼50% steps. As test accuracy reaches ∼66%, all optimizers start overfitting in this task; this is consistent with findings in prior work (Dahl et al., 2023). Since our update rule is normalized, it shows training norm dynamics consistent with AdamW. Moreover, VeLO is meta-trained with 200K unroll length on a large number of diverse tasks including ViTs and ImageNet dataset, whereas Celo2 is only meta-trained on small image MLP tasks (§4), which highlights its strong meta-generalization capability.

multiple updates in a single outer loop. This is much more computationally efficient than ES (used in VeLO) that does only a single optimizer update within an unroll; see Appendix D for full background on the meta-training setup. We use average loss from the unroll during meta-training as the meta-objective. Despite being meta-trained with average loss as the meta-objective, our optimizer yields strong final performance on large-scale tasks as demonstrated by experiments discussed next.

**Implementation details.** We conduct all large-scale evaluations in JAX (Bradbury et al., 2018) on a v4 TPU pod with 32 chips and 4 VM hosts, using fully sharded data parallelism (FSDP). For RL experiments, we use JAX (jit, pmap) to vectorize agent and environment computations end-to-end on device, avoiding TPU–CPU transfers. Our learned optimizer is meta-trained for 100K iterations with $K$=50 inner steps and 8 parallel tasks on Nvidia L40S GPU. Unless stated otherwise, optimizer evaluations sweep 7 learning rates logarithmically between 1e-3 and 1e-5, cosine decay schedule with linear warmup fraction 0.05 and report the best performing hyperparameter setting for each optimizer based on final performance (see §B for more details). VeLO is tested in default self-tuned mode by specifying target duration at initialization along with training loss at each step. Since Celo2 is a learned update rule, we implemented it in Optax (§C) and evaluated with a 1-line drop-in replacement in all the standard tasks used in this work. We used float32 as the default precision for language modeling experiments to decouple instabilities introduced by the learned optimizer experiments from mixed-precision instabilities during research. However, testing our update rule with bfloat16 on ImageNet ViT (Fig. 2) revealed no instability. We leave targeted study with respect to lower/mixed precisions to reduce memory footprint of learned optimizers for future work.

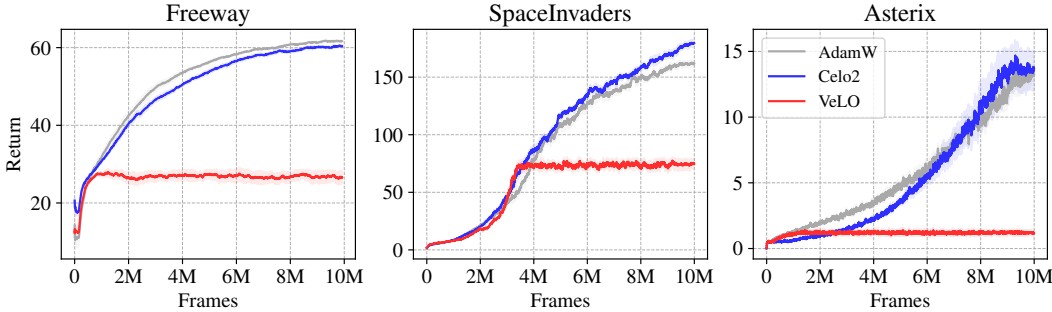

Figure 3: **Reinforcement Learning.** We directly evaluate our learned optimizer, Celo2, on Atari RL tasks using the PPO algorithm to learn the RL policy. Our results clearly indicate that Celo2 performs at par with a well-tuned AdamW baseline on these out-of-distribution tasks, while the VeLO baseline stagnates at a much lower return. The latter result can be corroborated by Figure 11 in Metz et al. (2022b).

## 4.1 RESULTS

**Generalizing to billion-scale models (GPT-3).** Although learned optimizers have been shown to generalize to small and medium-scale tasks (Metz et al., 2022b; Moudgil et al., 2025), they have fallen short of generalizing to large-scale pretraining tasks such as GPT (Achiam et al., 2023), which is highly important to the machine learning community. We show that Celo2, our proposed learned update rule, generalizes to GPT-3 1.3B, GPT-2 124M, and a 30M transformer pretraining tasks, see Fig. 1. All pretraining tasks follow the Chinchilla (Hoffmann et al., 2022) recommended data-to-model scaling ratio of 20, except GPT-3 1.3B, which uses a ratio of 10 due to compute constraints; this setup matches prior work (Marek et al., 2025). Notably, GPT-3 1.3B is $1,000,000\times$, or six orders of magnitude, larger than the tasks Celo2 is meta-trained on, so this is purely out-of-distribution performance. Not only does it scale gracefully and remain stable at this scale, it is also competitive with strong, well-tuned baselines such as VeLO and AdamW under the same tuning budget. We also found Celo2 to be competitive with Muon (Jordan et al., 2024), a recent state-of-the-art optimizer (Figure 7 in Appendix), despite being meta-trained on simple $8\times8$ image classification tasks (§4).

**Generalizing to longer unrolls.** In order to stress-test our learned update rule on long horizon tasks, we choose the ImageNet (Krizhevsky et al., 2012) classification task with the Vision Transformer (ViT) as our testbed. We perform unrolls for 50,000 steps; this is $25\times$ larger than the unrolls seen during meta-training. As a result, this is a strong test of length generalization for Celo2, exceeding previously attempted length generalization evaluations in literature (Moudgil et al., 2025; Thérien et al., 2024). Our results in Fig. 2 show that Celo2 not only generalizes reliably and stably to large-scale tasks, it also significantly outperforms VeLO and tuned AdamW baselines. Note that VeLO is optimized to achieve the lowest final validation loss; Celo2, on the other hand, achieves VeLO's target final validation loss $\sim2\times$ faster than VeLO. In addition, Celo2 surpasses 80% training accuracy much faster than the baselines within the given number of steps.

**Reinforcement learning.** This is another out-of-distribution generalization result from noisy (policy) gradients. Prior work (Metz et al., 2022a; Harrison et al., 2022; Moudgil et al., 2025; Thérien et al., 2024), with the exception of VeLO in learned optimizers, has not reported results on Reinforcement Learning tasks from a generalization perspective. However, such tasks are particularly valuable for evaluating generalization, as they involve long-horizon credit assignment, high variance in training dynamics, and non-stationary objectives, making them a challenging and informative benchmark for optimizer robustness. We reuse the hyperparameters for PPO and the environment from prior work (Lu et al., 2022). We further tune the AdamW LR using the search space suggested by prior work and our AdamW baseline closely matches the reported results. As a result, our AdamW baseline has near-optimal hyperparameters, for instance $\epsilon$=1e-5. Fig. 3 clearly indicates that Celo2, evaluated zero-shot on these RL tasks, outperforms VeLO by a significant margin, and is competitive with AdamW. This is despite it being meta-trained on $8\times8$ image classification tasks with potentially unfavorable hyperparameters for RL (for example, a low epsilon value). It is im-

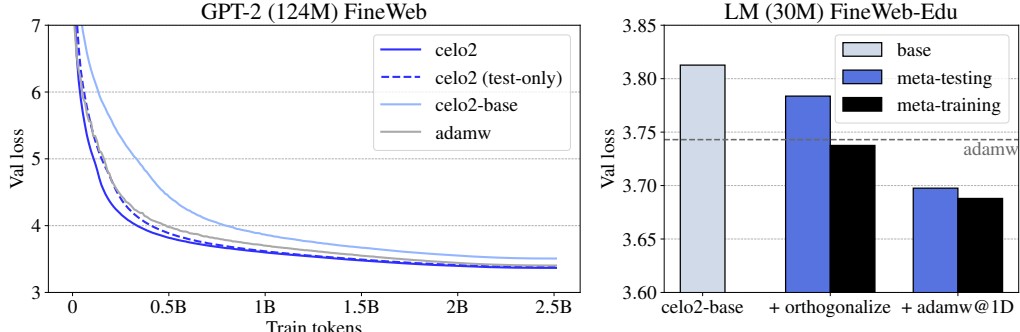

Figure 4: As shown in the figure on the left, Celo2-base that uses a simple learned MLP rule for all parameters without orthogonalization or AdamW, is able to scale stably on GPT-2 task. We find that both techniques (1) Orthogonalization and (2) Adam for 1D params improve performance when applied on top of Celo2-base. Applying these two techniques directly at test-time improves performance but meta-training with them is even better.

| hidden size | val loss↓ |
|---|---|
| 4 | 4.128 |
| 8 | **3.812** |
| 16 | 3.857 |
| 32 | 3.869 |

(a)

| RMS decay | val loss↓ |
|---|---|
| 0.999 | 3.893 |
| 0.95 | **3.812** |
| 0.99 | 3.865 |
| 0.9 | 3.922 |

(b)

| task aug. | val loss↓ |
|---|---|
|  | 4.417 |
| ✓ | **3.812** |

(c)

| RMS norm | | val loss↓ |
|---|---|---|
| meta-training | test-time | |
|  |  | 3.961 |
|  | ✓ | 3.992 |
| ✓ | ✓ | **3.812** |

(d)

| update form | val loss↓ |
|---|---|
| $\lambda_1 d$ | **3.812** |
| $\lambda_1 d \cdot e^{(\lambda_2 m)}$ | 3.900 |
| $\lambda_1 d \cdot e^{(\lambda_2 m_{\text{avg}})}$ | 4.075 |

(e)

Table 1: **Ablations over the ingredients of our proposed recipe.** We find that learned optimizer generalization performance is sensitive to architecture and meta-training hyperparameters and they must be ablated thoroughly to achieve strong performance. **(a) Hidden size:** learned update rule with hidden size 8 emerges as the ideal choice, resulting in a smaller yet more performant learned optimizer; **(b) RMS decay:** we find that a value of 0.95 for the RMS decay yields the best results; **(c) Task augmentation:** one of the ingredients in our proposed recipe is task augmentation; this result clearly shows a significant improvement in validation loss using task augmentation; **(d) RMS norm:** RMS norm is another component we propose for Celo2-base. Using RMS norm helps both during meta-testing and also meta-training. This trend is consistent with our overall results; **(e) Update form:** the best learned update rule predicts simply $d$ instead of more complex update rules that need to predict both $d$ and $m$. All ablations are evaluated on a 30M transformer decoder model trained on FineWeb-Edu with 600M tokens (§5). Celo2-base with our default settings are highlighted .

portant to note here that the VeLO paper also reported similar trends in the stagnation of reward and the failure of the model to learn after a point (refer to Figure 11 in Metz et al. (2022b)).

# 5 ABLATIONS

We now present ablations of each component in the Celo2 recipe. All experiments are on the LM (30M) task with the FineWeb-Edu dataset using 600M tokens, following the Chinchilla (Hoffmann et al., 2022) scaling law. The learning rate of the meta-learned rule is tuned on this task, and the best result for each ablation is reported in Table 1. Full hyperparameter details are in Appendix B.

**Orthogonalization.** Recent progress in optimization (Jordan et al., 2024), has proposed an optimizer that relies on orthogonalization to achieve faster convergence. Although this incurs a penalty in the form of wall clock time, the downstream benefits include lower final losses in addition to the already mentioned faster convergence. We find that Celo2 is highly compatible with orthogonalization, as demonstrated in Fig. 4. Specifically, adding orthogonalization to a base Celo2 model directly at meta-testing time yields an improvement over Celo2-base (referred to as *celo2 (test-only)* in Fig. 4). This improvement is amplified by adding orthogonalization while meta-training Celo2. We find that similar trends hold by using AdamW for 1-D / bias parameters, which is another recently adopted practice (Jordan et al., 2024). Fig. 4 clearly shows that adding orthogonalization to Celo2-base, followed by AdamW update for 1-D gives a compounding benefit in performance, both for meta-testing and meta-training. However, as shown in Fig. 4, our Celo2-base model which uses a *fully learned* update rule across all parameters is also highly stable on the GPT-2 (124M) task.

**Task augmentation.** Task augmentation is a useful technique (Metz et al., 2022b; Moudgil et al., 2025) to simulate a diverse range of tasks from a limited set of tasks during meta-training to enhance generalization. This is done by randomly sampling a scaling parameter and then rescaling the parameters of the optimizee network during meta-training. This leads to a perturbation in the gradients, thereby resulting in a higher coverage of the optimization landscape, analogous to the role that data augmentation plays in supervised learning. We find that task augmentation is critical for good performance (refer to Table 1(c) for task augmentation ablation). Qualitatively, we see that meta-training with task augmentation allows the use of a higher learning rate, thereby speeding up the optimization process and resulting in lower final validation loss.

**RMS normalization.** In order to achieve high performance by using a purely learned rule without any hand-designed update rules for 1-D parameters such as Adam, we find that RMS normalization plays a key role. The results of our ablations in Table 1(d) indicate that adding RMS normalization during the meta-training phase yields improved performance. This trend did not hold for using it simply during meta-testing with the Celo2-base update rule. In Appendix §A, we provide ablations on different normalization variants, showing that the choice in Alg. 1 performs best.

**Meta-training hyperparameters that matter.** In addition to the components discussed above, we find that generalization performance is sensitive to several architectural choices in the learned MLP update and to specific hyperparameters. The first is the *magnitude multiplier*: the exponential magnitude multiplier proposed by prior work (Metz et al., 2022a) does not improve meta-generalization (Table 1(e)). Using only the direction coefficient from the MLP update is sufficient. Averaging the exponential magnitude multiplier per tensor, as in Harrison et al. (2022), also reduces performance. The second factor is the *RMS decay* term with coefficient $\beta$, used similarly to Adam (see §D for background). For LM pretraining tasks, $\beta = 0.95$, the standard choice in recent LLM works Marek et al. (2025) gives the best generalization, and this is the default in Celo2. The third factor is the *MLP hidden size*: prior work (Metz et al., 2022b;a) uses a 2-layer MLP with 4 units per layer, but we find that using 8 units per layer provides the best balance between performance and efficiency.

**Runtime and memory overhead.** Celo2-base and Adam have identical wall clock time, since the optimizer update is an inexpensive part of the entire training step. However, in the number of parameters, Adam has a memory overhead of $3\times$ due to 2 additional (momentum and RMS decay) accumulators per-parameter. On the other hand, Celo2 has a memory overhead of $\sim 5\times$ since it maintains 3 momentum accumulators with 3 $\beta$ decays (unlike 1 in Adam), along with one RMS decay per-parameter and tensor-level Adafactor features (Metz et al., 2022b). Celo2 incurs a higher runtime cost ($1.3\times$) due to Newton-Schulz orthogonalization, though this may vary by task.

## 6 CONCLUSION

In this paper, we proposed a practically applicable, inexpensive, yet effective recipe for learned optimization. Our approach results in a win-win situation on several axes: stability, compute-efficiency, and performance. Our learned update rule has several benefits: 1) it is more compute efficient than prior work, taking merely 4.5 GPU hours for meta-training, 2) it shows strong out-of-distribution generalization across datasets, 3) it demonstrates powerful scaling on large models like GPT-3. We provide a detailed set of ablations to study the impact of each individual design decision proposed in our recipe. We hope that this work serves as an important milestone in the development and discovery of high-performing scalable learned optimizers.

## ACKNOWLEDGEMENTS

We acknowledge support from the FRQNT Doctoral (B2X) scholarship [AM], the FRQNT New Scholar and FRQNT-NOVA [EB] and resources provided by Compute Canada, Calcul Québec and Mila. We also extend our sincere gratitude to the Google TPU Research Cloud (TRC) program for generously providing TPU resources that made this research possible.

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

## A ADDITIONAL EXPERIMENTS AND ANALYSIS

**Normalization variants.** To achieve high meta-generalization performance, we explore several normalization strategies for the predicted updates. Table 2 compares different approaches: using raw MLP outputs, per-step RMS normalization, rolling RMS normalization (aggregated over time), and variants with hard clipping. Per-step RMS normalization (*norm*) achieves the best validation loss of 3.81268, substantially outperforming the raw output baseline (3.96168). Rolling normalization (*rolling norm*) also improves over the baseline but falls short of per-step normalization. Incorporating hard clipping, either with rolling normalization (*rolling norm clip*) or per-step normalization (*norm clip*), degrades performance compared to their non-clipped counterparts, suggesting that overly constraining update magnitudes can be detrimental. These results demonstrate that simple per-step RMS normalization is the most effective strategy for stabilizing meta-learned updates without requiring hand-designed rules.

|  | functional form | val loss↓ |
|---|---|---|
| raw output | $\Delta \boldsymbol{p}_t = \text{MLP}(...)$ | 3.961 |
| rolling norm | $\Delta \boldsymbol{p}_t / \sum^t \|\Delta \boldsymbol{p}_t\|_{\text{rms}}$ | 3.860 |
| rolling norm clip | $\Delta \boldsymbol{p}_t \cdot \min\left(1, \tau \cdot (\sum^t \|\Delta \boldsymbol{p}_t\|_{\text{rms}}) / \|\Delta \boldsymbol{p}_t\|_{\text{rms}}\right)$ | 3.999 |
| norm clip | $\Delta \boldsymbol{p}_t \cdot \min\left(1, \tau / \|\Delta \boldsymbol{p}_t\|_{\text{rms}}\right)$ | 4.084 |
| norm | $\Delta \boldsymbol{p}_t / \|\Delta \boldsymbol{p}_t\|_{\text{rms}}$ | **3.812** |

Table 2: We evaluate several ways of scaling the predicted update vector $\Delta \mathbf{p}_t$. "Raw output" applies the MLP output directly. "Rolling norm" normalizes by the cumulative RMS magnitude of past updates, while "rolling norm clip" additionally limits step size using a time-dependent threshold $\tau$. "Norm clip" clips each update based on its RMS norm. "Norm" simply normalizes each update by its own RMS magnitude and performs the best, achieving the lowest validation loss among all variants. All variants are evaluated on a language modeling task using a 30M-parameter Transformer trained on the FineWeb-Edu dataset with 600M tokens (§5).

**Stability and hyperparameter sensitivity.** As evident from the LM (30M) FineWeb-Edu plot on the right (Figure 5), Celo2 demonstrates stable optimization behavior across a range of learning rates, with performance characteristics that closely match standard optimizers like AdamW and VeLO. The best tuned AdamW and VeLO configurations are also plotted for reference from Figure 1. Learning rates toward the higher end of our sweep generally perform well, though the highest learning rate (0.001) underperforms compared to the best setting with a slightly lower learning rate (0.000464). This pattern, where moderately high learning rates optimize effectively while very high rates cause instability, is consistent across all our supervised learning experiments. We did not find Celo2's stability to be sensitive to any specific hyperparameter values within reasonable ranges. For reinforcement learning experiments, we present a parallel coordinate plot in Figure 6 showing sensitivity to ablated parameters in our sweep for the

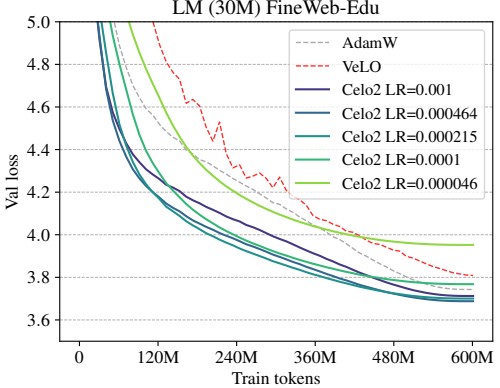

Figure 5: Validation loss curves for Celo2 with various learning rates uniformly sampled on log-scale between 1e-5 and 1e-3 on LM (30M) FineWeb-Edu dataset.

SpaceInvaders environment as a representative example. The highlighted yellow curves indicate high-return trajectories. Here too, Celo2's behavior aligns well with standard optimizers. The optimal learning rate value is close to the higher end of the range but not the highest, as the maximum learning rate leads to unstable optimization and consequently lower returns. The optimizer shows no particular sensitivity to random seed initialization, demonstrating robust performance across different experimental runs. We also ablate over the choice of linear learning rate decay (a common practice in these Atari RL tasks) and find that in this environment, Celo2 benefits from incorporating linear decay, though it maintains stable performance across both configurations.

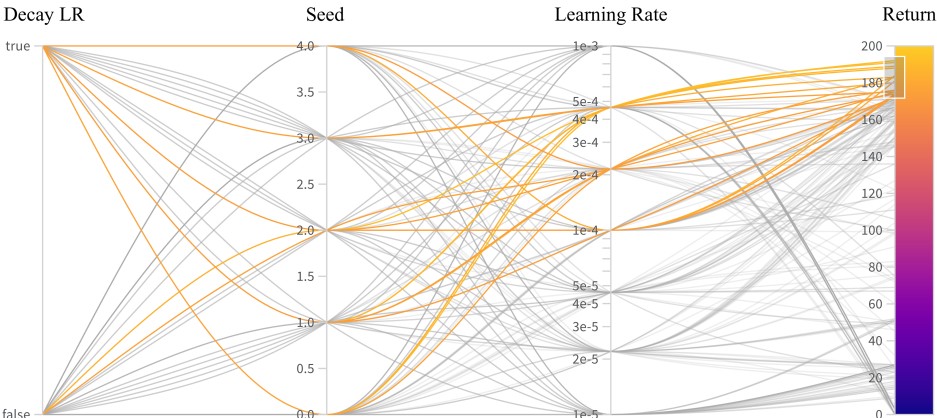

Figure 6: Parallel coordinate plot showing Celo2 hyperparameter sensitivity on SpaceInvaders RL task. Yellow curves indicate high-return configurations.

**Comparison with Muon.** We compare Celo2 with Muon (Jordan et al., 2024; Liu et al., 2025) on the GPT-2 (124M) FineWeb pretraining task in Figure 7. Muon is a hand-crafted optimizer for hidden layers in neural networks that orthogonalizes momentum with Newton-Schulz iterative process. For non-hidden parameters (embeddings, scalars), it uses Adam update rule. We use the Kimi variant (Liu et al., 2025), which applies RMS normalization to the orthogonalized update (as in Celo2) to match the RMS of Adam updates for 1D parameters. This enables a shared learning rate across embeddings and hidden layers and reduces tuning effort. Kimi-Muon uses an RMS multiplier of 0.2 for hidden layers to match the empirical RMS of Adam; we found this also works well for Celo2 and include it in the plot (Celo2 RMS×0.2 in Figure 7). Note that Celo2

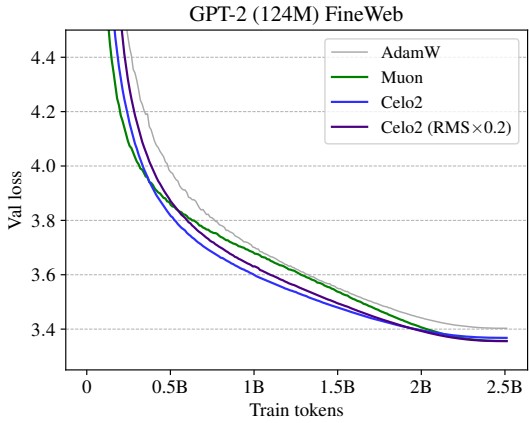

Figure 7: Comparison of Celo2 and Muon on GPT-2 (124M) FineWeb pretraining task.

RMS×0.2 variant and Muon only differ in the update rule on which Orthogonalization is applied: Muon uses momentum whereas Celo2 uses learned MLP update and rest of the optimizer update is identical since they both use RMS normalization with 0.2 multiplier and Adam for 1D parameters.

Overall, both Celo2 and Celo2 RMS×0.2 demonstrate competitive performance compared to Muon, achieving lower validation loss for majority of the run (Figure 7). In terms of final loss, Celo2 RMS×0.2 peforms better, achieving slightly lower validation loss (3.35588) than Muon (3.35636) while default Celo2 (without 0.2 RMS multiplier) achieves 3.36785. For this comparison, we tuned the Muon baseline by log-uniformly sampling 10 peak learning rates between 1e-4 and 1.0: (0.0001, 0.000215, 0.000464, 0.001, 0.00215, 0.00464, 0.01, 0.215, 0.464, 1.0). All optimizers use cosine decay schedule with warmup fraction 0.05, decay to zero and weight decay 0.1. Muon performs best at learning rate 0.01. For Celo2 variants, we use the same log-uniform sampling 7 learning rates between 1e-5 and 1e-3 as in our experiments in the main paper (§4): (0.00001, 0.0000215, 0.0000464, 0.0001, 0.000215, 0.000464, 0.001). Celo2 and Celo2 RMS×0.2 variant perform best at 0.000215 and 0.000464 learning rates, respectively. The learned MLP update rules tested in this study is meta-trained on simple 8×8 image classification tasks (§4); further meta-training or fine-tuning on language modeling tasks could improve performance, which we leave for future work.

# B  EXPERIMENTAL DETAILS

**Language Modeling.** For our large-scale language–modeling evaluations, we evaluate Celo2 on three autoregressive Transformer architectures ranging from 30M to 1.3B parameters, all of which were trained on a TPU pod. All models use the same decoder–only Transformer implementation provided in the public codebase[1] by Marek et al. (2025). Each model consists of token embeddings (input and output), $L$ pre-norm Transformer blocks, and a 2-layer MLP with inner dimension $F = 4D$; full model configs are presented in Table 3 below. Multi-head causal self-attention is implemented with query/key RMSNorm, RoPE positional embeddings, and either standard dot-product attention or TPU FlashAttention via `splash_attention_kernel` (automatically enabled when training on TPU). Weights are initialized using Xavier-uniform except embedding parameters, which are initialized using variance scaling with a normal distribution, fan-in mode, and a scaling factor of 1.0 along the output axis.

| model | D | L | F | H | N | T |
|---|---|---|---|---|---|---|
| 30M (LM-30M) | 384 | 6 | 1536 | 128 | 3 | 512 |
| 124M (GPT-2) | 768 | 12 | 3072 | 128 | 6 | 1024 |
| 1.3B (GPT-3) | 2048 | 24 | 8192 | 128 | 16 | 2048 |

D: hidden dimension, L: number of transformer layers,
F: feedforward inner dimension, H: attention head dimension,
N: number of attention heads, T: context/sequence length

Table 3: Language model architectures. We evaluate Celo2 on three autoregressive Transformer architectures. All models share the same architecture template but differ in size. $F = 4D$ and $N = D/H$. Vocab size for FineWeb dataset is 50257 and rounded to match device sharding requirements.

| hyperparameter | LM-30M | GPT-2 (124M) | GPT-3 (1.3B) |
|---|---|---|---|
| learning rate (celo2) | 0.000464 | 0.0002 | 0.00002 |
| learning rate (adamw) | 0.01976 | 0.0048 | 0.0002 |
| warmup fraction | 0.05 | 0.05 | 0.05 |
| batch size (global) | 256 | 512 | 512 |
| weight decay | 0.1 | 0.1 | 0.1 |
| adamw epsilon | 1e-8 | 1e-8 | 1e-8 |
| adamw $\beta_1$ | 0.9 | 0.9 | 0.9 |
| adamw $\beta_2$ | 0.999 | 0.999 | 0.999 |
| learning rate schedule | linear warmup + cosine decay | | |

Table 4: Hyperparameters used for the language modeling experiments in Figure 1.

Sharded parameters are placed across a 2D device mesh (`data × model`). Training follows standard next-token prediction using teacher forcing with cross-entropy loss. Large global batch sizes are supported through gradient accumulation. We evaluate all the optimizers by sweeping learning rate and weight decay hyperparameters. All training hyperparameters for tuned LM experiments from Figure 1 are described in Table 4. We search over learning rates and weight decay values for both Celo2 and AdamW. For Celo2, we sample seven learning rates logarithmically between 0.00001 and 0.001 (specifically: 1e-5, 2.15e-5, 4.6e-5, 1e-4, 2.15e-4, 4.64e-4, 1e-3) and test them with a linear warmup and cosine decay to zero schedule. For weight decay, we test three values: 0.0, 0.1, and 10.0. We include the high value of 10.0 because initial experiments with Celo2-base showed it can handle higher weight decay than typical optimizers. For AdamW, we consult tuned values / search space from prior work (Marek et al., 2025; Achiam et al., 2023), which were found using larger compute budgets than our Celo2 search (for example, 13 learning rates between 0.0002 and 0.03 in the LM-30M task for AdamW) and we further tune them with additional weight decay values discussed above. This makes AdamW a stronger baseline. Empirically, our search ranges appear well-chosen: for both optimizers, mid-range values generally perform best, with larger models preferring slightly lower learning rates.

---

[1] `https://github.com/martin-marek/batch-size`

**Image Classification.** Vision Transformer (ViT) experiments are conducted using the `jaxtransformer`[2], a minimal but performant Transformer repository implemented in JAX. Full hyperparameter config is presented in the Table 5 below. Models are trained on the ImageNet dataset with a base-size configuration (hidden size 768, 12 Transformer layers, 12 attention heads, MLP ratio 4). Images are preprocessed by resizing or padding the smaller side to make them square, then resized to 256×256 and passed as input to ViT model. During training, additional data augmentation is applied: images are randomly flipped horizontally, and a random resized crop (area range 5–100%, aspect ratio 0.75–1.33) is performed, followed by resizing back to 256×256. Pixel values are normalized to [-1, 1]. Training uses a batch size of 512 for a total of 50,000 steps. Weight decay is fixed to 0.1. ViTs are known to be data-hungry (Dosovitskiy et al., 2021; Zhai et al., 2022; Steiner et al., 2022; Touvron et al., 2021) and on this task begin to overfit around ∼66% validation accuracy, making stronger regularization crucial. We also tried lower weight values (0.01 and 0.001), but they consistently underperform compared to 0.1. The learning rate is swept log-uniformly from $1 \times 10^{-5}$ to $1 \times 10^{-3}$ for both Celo2 and AdamW optimizers. We found this to be an effective search space, as the optimal values for both optimizers lie within this range, with AdamW favoring comparatively higher learning rates. Fully Sharded Data Parallel (FSDP) is leveraged to efficiently distribute model parameters and optimizer states across multiple devices, enabling large-batch training on TPUs.

| hyperparameter | value |
|---|---|
| learning rate (celo2) | 0.000046 |
| learning rate (adamw) | 0.0001 |
| dataset | imagenet-256 |
| model | ViT |
| hidden size | 768 |
| depth (layers) | 12 |
| number of heads | 12 |
| MLP expand ratio | 4 |
| weight decay | 0.1 |
| batch size (global) | 512 |
| training steps | 50,000 |
| warmup fraction | 0.02 |
| learning rate schedule | linear warmup + cosine decay |

Table 5: Hyperparameters used for the Vision Transformer (ViT) experiments in Figure 2.

**Reinforcement Learning.** We conduct reinforcement learning experiments using a distributed PPO-style framework[3] implemented in JAX. Agents are trained on multiple Atari environments, including Asterix, Freeway, and Space Invaders. The hyperparameter configurations used to generate the results in Figure 3 are listed in Table 6. Each environment runs in parallel across multiple actors, and observations are flattened and logged for stability. Policies are parameterized as actor-critic networks with two hidden layers of size 64 and tanh activations, producing action distributions and value estimates. Training proceeds in mini-batches collected over multiple steps per environment. Following common practices and PPO hyperparameters from the `purejaxrl` repository, we perform multiple PPO update epochs per batch of trajectories, subdivide batches into minibatches for gradient updates, and use generalized advantage estimation (GAE) with discount factor $\gamma = 0.99$ and GAE parameter $\lambda = 0.95$. The PPO objective includes clipped surrogate losses, with clip epsilon defining the maximum allowed deviation. Entropy regularization encourages exploration, weighted by the entropy coefficient. Similar to other experiments, we sweep learning rates log-uniformly from $1 \times 10^{-5}$ to $1 \times 10^{-3}$ for AdamW and Celo2. Gradients are clipped to a maximum norm, and learning rates are either kept constant or linearly decayed to zero (we found linear decay coupled with high learning rate to be slightly better) which is a standard practice in the aforementioned `purejaxrl` repository that multiple works build on (Lu et al., 2022; Sapora et al., 2024).

---

[2]https://github.com/kvfrans/jaxtransformer
[3]https://github.com/luchris429/purejaxrl

| hyperparameter | freeway | space-invaders | asterix |
|---|---|---|---|
| learning rate (celo2) | 0.000215 | 0.000464 | 0.0001 |
| learning rate (adamw) | 0.001 | 0.001 | 0.001 |
| total timesteps | 10M | 10M | 10M |
| discount factor ($\gamma$) | 0.99 | 0.99 | 0.99 |
| GAE parameter ($\lambda$) | 0.95 | 0.95 | 0.95 |
| num steps per update | 128 | 128 | 128 |
| parallel actors | 64 | 64 | 64 |
| PPO clip epsilon | 0.2 | 0.2 | 0.2 |
| PPO entropy coefficient | 0.01 | 0.01 | 0.01 |
| gradient norm clipping | 0.5 | 0.5 | 0.5 |
| value loss coefficient | 0.5 | 0.5 | 0.5 |
| random seeds | 5 | 5 | 5 |
| weight decay | 0.0 | 0.0 | 0.0 |
| adamw epsilon | 1e-5 | 1e-5 | 1e-5 |
| adamw $\beta_1$ | 0.9 | 0.9 | 0.9 |
| adamw $\beta_2$ | 0.999 | 0.999 | 0.999 |
| learning rate schedule | linear decay | linear decay | linear decay |

Table 6: Hyperparameters used for the RL experiments in Figure 3.

## C  JAX IMPLEMENTATION

The complete JAX implementation of Celo2 is available at: `https://github.com/amoudgl/celo2`. In this section, we provide a high-level overview of the implementation and illustrate key design choices using simplified code snippets and pseudocode. The goal is to make the core ideas accessible to readers who may be new to JAX and Optax, while keeping the presentation focused on the conceptual structure rather than low-level implementation details.

**Optax overview.** Celo2 is implemented in JAX with Optax (Bradbury et al., 2018) syntax. Specifically, in Optax, each optimizer is expressed as a *transformation* that maps parameters and gradients to updated parameters together with any auxiliary state. For example, the standard Adam optimizer is constructed via `optax.adam(learning_rate)`, which returns a transformation consisting of an `init` function that initializes the optimizer state and an `update` function that applies the Adam update rule. Concretely, in python, an optimization step can be written as:

```python
# for example, adam transformation is initialized as:
tx = optax.adam(learning_rate)

# initialize optimizer state
state = tx.init(params)

# update state
updates, state = tx.update(grads, state, params)
params = optax.apply_updates(params, updates)
```

Celo2 follows this same abstraction: we implement our method as a custom Optax transformation with its own state definition, initialization logic, and update rule, ensuring compatibility with JAX's functional style and seamless integration with existing Optax training pipelines. One of the powerful features of this transformation view is that transformations can be composed using `optax.chain`. Multiple transformations, such as gradient clipping, weight decay, and learning rate scaling, can be appended in a modular way. For example:

```python
# Compose multiple transformations using optax.chain
tx = optax.chain(
    celo2_optax(...),                           # celo2 transform function
    optax.add_decayed_weights(0.1),             # weight decay
    optax.scale_by_learning_rate(learning_rate) # scale by learning rate (-lr*update)
)

state = tx.init(params)
updates, state = tx.update(grads, state, params)
```

```
params = optax.apply_updates(params, updates)
```

This modular design allows us to implement learned update rules such as Celo2 cleanly, while keeping each component of the update logic separated and reusable within standard Optax pipelines. In addition to modular transformations, Optax also supports flexible learning rate schedules that can be combined seamlessly with any optimizer. For instance, one can define a schedule such as cosine decay, exponential decay and then scale updates accordingly:

```
from optax.schedules import warmup_cosine_decay_schedule

# Define a cosine learning rate schedule
schedule = warmup_cosine_decay_schedule(init_lr, peak_lr, warmup_steps, num_opt_steps,
    end_lr)

# Apply schedule to Celo2 transformation
tx = optax.chain(
    celo2_optax(...),
    optax.scale_by_learning_rate(schedule)  # dynamically scales updates with schedule
)
```

Furthermore, Optax provides the `optax.multi_transform` utility to apply different update rules to different subsets of parameters. This is particularly useful when one wants to treat 1D parameters (e.g., embeddings, biases, layer norms) differently from 2D+ parameters (e.g., weight matrices, convolution kernels). With `optax.multi_transform`, separate transformations can be defined for each parameter group and combined into a single optimizer:

```
# Define transformations for 1D and 2D+ parameters
tx_1d = optax.chain(optax.adam(...), ...)
tx_2d = optax.chain(celo2_optax(...), ...)

# Create param_labels as a function that returns a label pytree
param_labels = lambda params: jax.tree_map(
    lambda p: '1d' if p.ndim == 1 else '2d',
    params
)

# Combine using multi_transform
tx = optax.multi_transform(
    transforms={'1d': tx_1d, '2d': tx_2d},
    param_labels=param_labels
)

# Initialize and use
state = tx.init(params)
updates, state = tx.update(grads, state, params)
params = optax.apply_updates(params, updates)
```

**Celo2 optax transformation.** The Celo2 transformation (`CeloOptaxTransformation`) is implemented in JAX as a custom `GradientTransformation` following the standard Optax interface. Its internal state (`CeloOptaxState`) consists of rolling averages for momentum, RMS, and Adafactor-style scaling, along with a step counter. At each update, the rolling statistics are updated, and per-parameter updates are computed by applying a small MLP defined in the Celo2 class.

```
import functools
from typing import Optional
import flax
import jax
import jax.numpy as jnp
import chex
import optax

@flax.struct.dataclass
class CeloOptaxState:
    """Internal state of the Celo2 optimizer."""
    rms_rolling: chex.ArrayTree
    mom_rolling: chex.ArrayTree
    fac_rolling: chex.ArrayTree
    step: jnp.ndarray
```

```python
class CeloOptaxTransformation(optax.GradientTransformation):
    """
    Optax-compatible gradient transformation for Celo2.
    Computes per-parameter updates using rolling statistics and a small MLP.
    """

    def __init__(self, celo2, theta: dict):
        """
        Args:
            celo2: Instance of the Celo2 optimizer
            theta: Meta-parameters or pretrained state for the MLP
        """
        self.celo2 = celo2
        self.theta = theta

    def init(self, params: chex.ArrayTree) -> CeloOptaxState:
        """Initialize rolling statistics and step counter."""
        mom_acc, rms_acc, fac_acc = self.celo2.accumulators_for_decays()
        return CeloOptaxState(
            mom_rolling=mom_acc.init(params),
            rms_rolling=rms_acc.init(params),
            fac_rolling=fac_acc.init(params),
            step=jnp.asarray(0, dtype=jnp.int32),
        )

    def update(
        self,
        grads: chex.ArrayTree,
        state: CeloOptaxState,
        params: Optional[chex.ArrayTree] = None,
    ) -> tuple[chex.ArrayTree, CeloOptaxState]:
        """Compute per-parameter updates and return new optimizer state."""
        # Increment step and clip gradients as in Celo/VeLO
        step = optax.safe_increment(state.step)
        grads = jax.tree_util.tree_map(lambda g: jnp.clip(g, -1000.0, 1000.0), grads)

        # Update rolling statistics
        mom_acc, rms_acc, fac_acc = self.celo2.accumulators_for_decays()
        next_mom = mom_acc.update(state.mom_rolling, grads)
        next_rms = rms_acc.update(state.rms_rolling, grads)
        next_fac, fac_g = fac_acc.update(state.fac_rolling, grads)

        # Prepare MLP function
        apply_mlp = functools.partial(self.celo2.mlp_apply, self.theta)

        # Compute per-parameter updates
        updates = jax.tree_util.tree_map(
            apply_mlp,
            params,
            grads,
            next_mom.m,
            next_rms.rms,
            fac_g,
            next_fac.v_col,
            next_fac.v_row,
            next_fac.v_diag
        )

        # Update optimizer state
        new_state = CeloOptaxState(
            mom_rolling=next_mom,
            rms_rolling=next_rms,
            fac_rolling=next_fac,
            step=step,
        )
        return updates, new_state
```

The above Celo2 transformation can be directly plugged in a standard Optax chain. Following example constructs a simple Celo2 optimizer with weight decay and learning rate scaling as in `celo2-base` variant:

```python
def create_celo2_optimizer(params, learning_rate, weight_decay=0.0, checkpoint_path=None):
    """Create a Celo2 optimizer with weight decay."""
    celo2 = Celo2(...)
    theta = celo2.init_params()
    pretrained_state = load_state(checkpoint_path, theta) if checkpoint_path else None

    return optax.chain(
        CeloOptaxTransformation(celo2, pretrained_state),
        optax.add_decayed_weights(weight_decay),
        optax.scale_by_learning_rate(learning_rate)
    )
```

For advanced usage, different update rules can be applied to different parameter groups using `optax.multi_transform` as described in optax overview section above; for example Adam for embeddings and Celo2 for other parameters (2D+ tensors) as in language modeling experiments (§4)).

Notice that the above `CeloOptaxTransformation` implementation relies on an instance of the `Celo2` optimizer class. This object is responsible for maintaining and updating the internal rolling statistics (momentum, RMS, and Adafactor-style scaling), providing the accumulators for these decays, and defining the MLP function that computes per-parameter updates. In essence, the Optax transformation acts as a wrapper that translates standard gradient inputs into updates generated by the `Celo2` optimizer logic which is described next.

**Celo2 optimizer.** The `Celo2` class below implements a learned optimizer that tracks rolling statistics for momentum, RMS, and Adafactor-style accumulators, and computes per-parameter updates through a small MLP with normalized inputs and outputs. Optionally, updates can be orthogonalized using the Newton–Schulz method. The following pseudocode shows the adapted `Celo2` class, based on the original implementation (available here), along with an example of its usage.

```python
import jax
import jax.numpy as jnp
import flax
import haiku as hk
import optax
import functools
from learned_optimization.learned_optimizers import common

def orthogonalize_via_newton_schulz(x, ns_coeffs, ns_steps=5, eps=1e-8):
    if x.ndim < 2:
        raise ValueError("Input must have >= 2 dims")
    transposed = False
    if x.shape[-2] > x.shape[-1]:
        x = jnp.swapaxes(x, -2, -1)
        transposed = True
    x /= (jnp.linalg.norm(x, axis=(-2, -1), keepdims=True) + eps)
    def ns_iter(_, x):
        x_mT = jnp.swapaxes(x, -2, -1)
        a = x @ x_mT
        b = ns_coeffs[1] * a + ns_coeffs[2] * a @ a
        return ns_coeffs[0] * x + b @ x
    x = jax.lax.fori_loop(0, ns_steps, ns_iter, x)
    if transposed:
        x = jnp.swapaxes(x, -2, -1)
    return x

class Celo2:
    def __init__(self,
                 ff_hidden_size=8,
                 ff_hidden_layers=2,
                 momentum_decays=(0.9, 0.99, 0.999),
                 rms_decays=(0.95,),
                 adafactor_decays=(0.9, 0.99, 0.999),
                 orthogonalize=True,
                 ns_coeffs=(3.4445, -4.7750, 2.0315),
                 ns_iters=5,
```

```python
                ns_eps=1e-8):
        self.ff_hidden_size = ff_hidden_size
        self.ff_hidden_layers = ff_hidden_layers
        self.momentum_decays = jnp.asarray(momentum_decays)
        self.rms_decays = jnp.asarray(rms_decays)
        self.adafactor_decays = jnp.asarray(adafactor_decays)
        self.orthogonalize = orthogonalize
        self.ns_coeffs = jnp.asarray(ns_coeffs)
        self.ns_iters = ns_iters
        self.ns_eps = ns_eps
        self.act_fn = jax.nn.relu

        # Haiku transform for per-parameter MLP
        self.apply_mlp = hk.without_apply_rng(hk.transform(self._apply_mlp))

    def accumulators_for_decays(self):
        """
        Returns rolling statistics for momentum, RMS, and Adafactor-style scaling.
        """
        mom_acc = common.vec_rolling_mom(self.momentum_decays)
        rms_acc = common.vec_rolling_rms(self.rms_decays)
        fac_acc = common.vec_factored_rolling(self.adafactor_decays)
        return mom_acc, rms_acc, fac_acc

    def _second_moment_normalizer(self, x, axis, eps=1e-9):
        rms = jnp.mean(jnp.square(x), axis=axis, keepdims=True)
        return x * jax.lax.rsqrt(rms + eps)

    def _apply_mlp(self, param, grad, mom, rms, fac_g, fac_vec_col, fac_vec_row, fac_vec_v,
     summary_prefix):
        """
        Per-parameter MLP that computes updates.
        """
        inps = [grad, param, mom, rms, fac_g, fac_vec_col, fac_vec_row, fac_vec_v]
        axis = list(range(len(param.shape)))[-2:]

        # Input normalization
        inps = [self._second_moment_normalizer(x, axis=axis) for x in inps]

        # Concatenate features and apply MLP
        x = jnp.concatenate([jnp.reshape(i, (-1, i.shape[-1])) for i in inps], axis=-1)
        for _ in range(self.ff_hidden_layers):
            x = hk.Linear(self.ff_hidden_size)(x)
            x = self.act_fn(x)
        out = hk.Linear(1)(x)
        step = jnp.reshape(out, param.shape)

        # Newton-Schulz orthogonalization
        if self.orthogonalize and step.ndim >= 2:
            step = orthogonalize_via_newton_schulz(step, self.ns_coeffs, self.ns_iters, self
    .ns_eps)

        # Output normalization
        step = self._second_moment_normalizer(step, axis=axis)
        return step

# --- Example initialization --- #
celo2 = Celo2(
    ff_hidden_size=8,
    ff_hidden_layers=2,
    momentum_decays=(0.9, 0.99, 0.999),
    rms_decays=(0.95,),
    adafactor_decays=(0.9, 0.99, 0.999),
    orthogonalize=True,
)
lr_schedule = warmup_cosine_decay_schedule(init_lr, peak_lr, warmup_steps, num_opt_steps)
celo2_opt = optax.chain(
    CeloOptaxTransformation(celo2, pretrained_opt_params),
    optax.add_decayed_weights(weight_decay),
    optax.scale_by_learning_rate(lr_schedule)
)
```

```python
scaled_lr_schedule = lambda step: embedding_lr_mult * lr_schedule(step)
adam_opt = optax.adamw(scaled_lr_schedule, b1=0.9, b2=0.95, weight_decay=weight_decay)

# Combine using multi_transform, labeling embedding parameters as 'adam' and others as '
    celo2'
tx = optax.multi_transform(
    transforms={
        'celo2': celo2_opt,
        'adam': adam_opt
    },
    param_labels=lambda params: jax.tree.map_with_path(
        lambda path, val: 'adam' if 'embed' in jax.tree_util.keystr(path) else 'celo2',
        params
    )
)

# Use just like a standard optax optimizer!
updates, state = tx.update(grads, state, params)
```

## D  ADDITIONAL BACKGROUND

This section provides background on learned optimizers for readers new to the field. We focus on two key components: the architectural design of MLP-based learned optimizers and the meta-training procedure used to train them. For the architecture, we describe the Adafactor MLP learned optimizer introduced by Metz et al. (2022a), which serves as the foundation for several state-of-the-art learned optimizers including VeLO (Metz et al., 2022b) and Celo2, proposed in this work. For meta-training, we explain the bi-level optimization framework and describe key techniques that enable practical meta-training of learned optimizers: Persistent Evolution Strategies (PES) (Vicol et al., 2021), an efficient gradient estimation method, and task augmentation (Metz et al., 2022b; Moudgil et al., 2025), which improves generalization to unseen tasks. Finally, we present the complete outer-training loop that integrates all these components.

**Learned MLP update.** Several learned optimizer architectures have been proposed in previous works (Andrychowicz et al., 2016; Metz et al., 2020; Almeida et al., 2021; Metz et al., 2022a;b), each varying in the types of inputs they process, the elements they maintain within their state, and the functional form of their outputs. In this section, we review a learned optimizer parameterized using a multi-layer perceptron (MLP) (Metz et al., 2022a) (Adafactor MLP LOpt) that our work and other state-of-the-art works like VeLO (Metz et al., 2022b) build upon. It is a simple MLP-based learned optimizer that can serve as a drop-in replacement for hand-designed update rules such as SGD or Adam (Kingma & Ba, 2015).

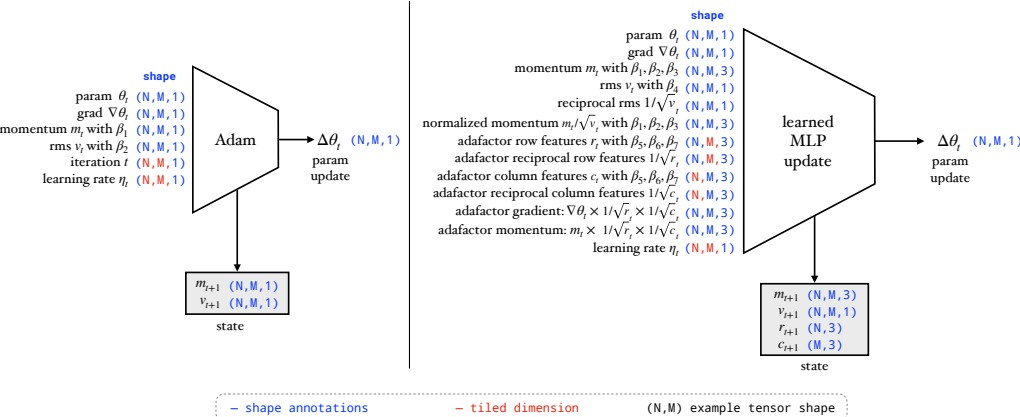

Figure 8: **Comparing Adam with learned MLP update rule.** Given a parameter tensor of shape $(N, M)$, the Adam update (left) maintains two additional accumulators per parameter, leading to a $3\times$ memory overhead over SGD. In contrast, the learned MLP optimizer proposed by Metz et al. (2022a) (right) incorporates additional momentum, root mean square-normalized features, and Adafactor-style row and column features. While the MLP optimizer maintains extra row and column accumulators, their memory cost scales *sublinearly* with the total parameter count $N \times M$. In total, MLP consists of <200 params (Metz et al., 2022a) and provides an effective trade-off between runtime and memory overhead, making it practical for large-scale tasks. We adapt the same learned MLP optimizer illustrated on the right in our work for per-parameter updates (with a few changes, see § 5). Tensor shape annotations are shown in blue, with tiled dimensions in red.

Concretely, at a given iteration $t$, the Adafactor MLP learned optimizer takes as input the parameter vector $\boldsymbol{\theta}_t$ and gradient $\nabla\boldsymbol{\theta}_t$. These values are used to build per-parameter input features that are passed to the learned MLP, as illustrated in Figure 8. The learned MLP takes these input features and updates different accumulators in its state $\boldsymbol{s}_t$, which we describe in detail next.

The Adafactor MLP optimizer maintains four types of accumulators for each tensor (i.e., layer weight or bias) in its state, as illustrated in Figure 8: (1) gradient momentum $m_t$ with $\beta_1, \beta_2, \beta_3$, (2) root mean square gradient accumulator $v_t$ with $\beta_4$, (3) Adafactor row feature accumulator $r_t$ with $\beta_5, \beta_6, \beta_7$, and (4) Adafactor column feature accumulator $c_t$ with $\beta_5, \beta_6, \beta_7$. The momentum $m_t$ and RMS $v_t$ accumulators are "per-parameter," meaning they correspond to each parameter in a given tensor. In contrast, the Adafactor accumulators $r_t$ and $c_t$ are "per-tensor," as these features

are extracted from the entire parameter tensor. Formally, given a parameter $\theta$ and its gradient $\nabla\theta$ from parameter tensor $\boldsymbol{\theta}_t \in \mathbb{R}^{N \times M}$, momentum $m_t$ and RMS accumulators $v_t$ for each parameter are updated as follows:

$$
\begin{aligned}
m_{t+1} &= \beta_i m_t + (1 - \beta_i)\nabla\theta \quad \forall i \in \{1, 2, 3\} \\
v_{t+1} &= \beta_4 v_t + (1 - \beta_4)\nabla\theta^2
\end{aligned}
\tag{1}
$$

where $\beta_1, \ldots, \beta_4$ are scalar constants. In Metz et al. (2022a;b), three momentum accumulators are used with $\beta_1, \beta_2, \beta_3 = (0.9, 0.99, 0.999)$ and one RMS accumulator with $\beta_4 = 0.999$. In Celo2, we found $\beta_4 = 0.95$ to be optimal for LM pretraining tasks, see §5. Adafactor accumulators $(r_t, c_t)$ maintain row and column mean of squared gradient tensor in each iteration:

$$
\begin{aligned}
r_{t+1} &= \beta_j r_t + (1 - \beta_j)\text{row-mean}(\nabla\boldsymbol{\theta}^2) \quad \forall j \in \{5, 6, 7\} \\
c_{t+1} &= \beta_j c_t + (1 - \beta_j)\text{col-mean}(\nabla\boldsymbol{\theta}^2)
\end{aligned}
\tag{2}
$$

where $\beta_5, \beta_6, \beta_7$ are scalar constants set to $(0.9, 0.99, 0.999)$ in Metz et al. (2022b;a). These row and column accumulators are tiled (replicated) corresponding to each parameter position and are then used to compute Adafactor momentum $m_t \times 1/\sqrt{r_t} \times 1/\sqrt{c_t}$ and Adafactor gradient $\nabla\theta \times 1/\sqrt{r_t} \times 1/\sqrt{c_t}$ features, as illustrated in Figure 8.

These accumulated values are then used to build input features for each parameter, which are passed to the learned MLP rule. Metz et al. (2022a) also passes global training progress features $\boldsymbol{\omega}^p$ as input to the MLP update rule, which we omit since we defer step-size tuning to the practitioner (§3). Concretely, Metz et al. (2022a) constructs training progress features $\boldsymbol{\omega}^p$ are computed by taking the current iteration $t$ and computing $\tanh(t/x)$ where $x \in \{1, 3, 10, 30, 100, 300, 1000, 3000, 10k, 30k, 100k\}$. In total, the learned MLP optimizer by Metz et al. (2022a) takes 39 input features as input for each parameter. These features are processed through a 2-layer MLP network with 4 hidden units each and ReLU activations (we ablate over hidden size in §1), which has total 197 parameters (Metz et al., 2022a). The forward pass through the learned MLP returns two outputs of the same dimensionality as $\boldsymbol{\theta}_t$, corresponding to direction $\boldsymbol{d}$ and magnitude $\boldsymbol{m}$, which are used to compute parameter updates $\Delta\boldsymbol{\theta}_t$ and then parameters $\boldsymbol{\theta}_{t+1}$ as follows:

$$
\begin{aligned}
\Delta\boldsymbol{\theta}_t &= \lambda_1 \boldsymbol{d} \cdot e^{(\lambda_2 \boldsymbol{m})} \\
\boldsymbol{\theta}_{t+1} &= \boldsymbol{\theta}_t - \Delta\boldsymbol{\theta}_t,
\end{aligned}
\tag{3}
$$

where $\lambda_1$ and $\lambda_2$ are fixed scalars set to low values (0.001) to keep meta-training stable. The learned optimizers discussed in this work, including VeLO (Metz et al., 2022a), Celo (Moudgil et al., 2025), and our proposed Celo2, roughly maintain the same Adafactor MLP architecture but differ in minor details such as accumulator constants ($\beta$ values), progress features and update functional form. We refer the reader to the respective works for more details.

**Meta-training problem.** Unlike hand-designed optimizers, learned optimizers' parameterized update function $\phi$ is trained through a meta-training process. A standard approach to meta-training involves solving a bi-level optimization problem that consists of an inner problem, which optimizes network parameters $\boldsymbol{\theta}$ using the learned optimizer update $\phi$ on a sampled task, and an outer problem, which optimizes the learned optimizer parameters $\phi$ based on feedback from the inner loop (Andrychowicz et al., 2016; Wichrowska et al., 2017; Metz et al., 2019a; 2020; Vicol et al., 2021). Formally, given a set of optimization tasks $\mathcal{T}$, the learned optimizer parameters $\phi$ are obtained by sampling an optimization task consisting of data distribution $\mathcal{D}$, initial network parameters $\boldsymbol{\theta}_0$, and a training objective $\mathcal{L}$, and solving the bi-level problem below:

$$
\phi^* = \arg\min_{\phi} \mathbb{E}_{(\mathcal{D}, \mathcal{L}, \boldsymbol{\theta}_0) \sim \mathcal{T}} \mathbb{E}_{X_t \sim \mathcal{D}}\left(\frac{1}{T}\sum_{t=0}^{T-1} \mathcal{L}(X_t; \boldsymbol{\theta}_t, \phi)\right),
\tag{4}
$$

where the inner loop is recursively defined for $t = [0, T-1]$ as:

$$
\begin{aligned}
(\boldsymbol{\theta}_t, \boldsymbol{s}_t) &= (\boldsymbol{\theta}_0, \boldsymbol{0}) && \text{if } t = 0, & (5) \\
(\boldsymbol{\theta}_t, \boldsymbol{s}_t) &= f_\phi(\boldsymbol{\theta}_{t-1}, \nabla\boldsymbol{\theta}_{t-1}, \mathcal{M}_{t-1}, \boldsymbol{s}_{t-1}) && \text{if } t > 0; & (6) \\
\nabla\boldsymbol{\theta}_t &= \frac{\partial\mathcal{L}(X_t; \boldsymbol{\theta}_t, \phi)}{\partial\boldsymbol{\theta}_t}; \quad X_t \sim \mathcal{D} && \forall t. & (7)
\end{aligned}
$$

Here $T$ denotes the unroll length in the inner loop and $X$ denotes sampled data from $\mathcal{D}$. The outer training objective or meta-objective is based on the mean loss as formalized above or, less commonly, the final loss of the inner loop (Metz et al., 2019a; 2020; 2022b). Computing meta-gradients for $\phi$ in Eq. 4 is challenging, as we explain next.

**Evolution strategies for gradient estimation.** Directly back-propagating through the inner loop in Eq. 4 to compute meta-gradients can lead to noisy gradient estimates, especially when the unroll length $T$ is large, due to the accumulation of errors through the computational graph (Metz et al., 2019a). An alternative approach is to use Evolution Strategies (ES) (Rechenberg, 1973; Hansen & Ostermeier, 2001; Wierstra et al., 2014; Salimans et al., 2017), a class of black-box optimization methods that estimate gradients by evaluating the objective function at perturbed parameter values. ES estimates the gradient of a Gaussian-smoothed objective with respect to parameters $\phi$ by sampling perturbations $\epsilon \sim \mathcal{N}(\mathbf{0}, \sigma^2 \mathbf{I})$ and evaluating:

$$\nabla_{\phi} J(\phi) \approx \frac{1}{n\sigma^2} \sum_{i=1}^{n} J(\phi + \epsilon_i)\epsilon_i, \tag{8}$$

where $n$ is the number of perturbation samples and $\sigma$ is the perturbation scale. This gradient estimate is obtained by evaluating the objective function $J$ at $n$ different perturbed versions of $\phi$ (specifically at $\phi + \epsilon_i$ for each $i$), and then computing a weighted average of the perturbations based on how much each perturbation improved or worsened the objective. ES has the advantage of not requiring backpropagation through the inner loop, making it memory-efficient and applicable even when the inner optimization process is non-differentiable or extremely long. However, standard ES applied to the full unroll of length $T$ in Eq. 4 can be computationally expensive, as it requires completing entire training runs for each perturbation sample before computing any meta-gradients. This is where Persistent Evolution Strategies (PES) provides a practical solution.

**Persistent Evolution Strategies (PES).** PES (Vicol et al., 2021) extends ES to work efficiently with truncated unrolls while maintaining unbiased gradient estimates. Instead of unrolling the entire inner loop of length $T$ before computing meta-gradients, PES divides the inner loop into shorter segments of length $K$ (where $K \ll T$). After each segment, meta-gradients are computed using ES on that truncated segment, and the learned optimizer parameters $\phi$ are updated without resetting the inner optimization. The key insight of PES is that it maintains persistence across truncations. Specifically, the optimizer state $s_t$ and inner problem parameters $\theta_t$ are carried forward after each meta-gradient update, and perturbations to the learned optimizer parameters are accumulated over time rather than reset. This persistence is crucial: without it, truncated gradient estimates would be biased because they would not account for how parameter updates affect future optimization trajectories. PES provides unbiased gradient estimates in the truncated setting by carefully accounting for how perturbations $\epsilon$ to the learned optimizer parameters affect both the immediate truncated objective and the carry-over effects through the persistent state.

Formally, for a truncated segment from step $t$ to $t + K$, PES estimates:

$$\nabla_{\phi} J_{\text{truncated}} \approx \frac{1}{n\sigma^2} \sum_{i=1}^{n} \left( \sum_{j=t}^{t+K-1} \mathcal{L}(X_j; \theta_j^{(i)}, \phi + \xi_j^{(i)}) \right) \xi_j^{(i)}, \tag{9}$$

where $\xi_j^{(i)} = \sum_{\tau \leqslant j} \epsilon_{\tau}^{(i)}$ denotes the accumulated perturbation for particle $i$, and $\theta_j^{(i)}$ denotes the inner parameters at step $j$ when using the perturbed optimizer, starting from the persistent state at step $t$. This makes meta-training significantly more efficient, as the learned optimizer can be updated frequently (every $K$ steps) rather than only after full unrolls of length $T$. The frequent updates also lead to better training dynamics, as the learned optimizer can adapt more quickly to improve performance. We use PES for computing meta-gradients in our work, following prior work (Metz et al., 2022a; Harrison et al., 2022; Moudgil et al., 2025).

**Task augmentation.** To improve the generalization of learned optimizers, meta-training can incorporate task augmentation (Metz et al., 2022b; Moudgil et al., 2025). Task augmentation modifies the sampled optimization tasks during meta-training to create a more diverse training distribution, helping the learned optimizer generalize better to unseen tasks at test time. One effective form of augmentation is parameter reparameterization, where the initial parameters $\theta_0$ of each sampled task are scaled by random factors. This reparameterization can be done at three levels of granularity: (1) *global*, where all parameters are scaled by the same factor, (2) *per-tensor*, where each weight matrix or bias vector is scaled independently, or (3) *per-parameter*, where each individual parameter is

scaled independently. Formally, given a sampled task with initial parameters $\boldsymbol{\theta}_0$, the reparameterized initialization is $\tilde{\boldsymbol{\theta}}_0 = \boldsymbol{\alpha} \odot \boldsymbol{\theta}_0$, where $\boldsymbol{\alpha}$ is a scaling factor (or tree of scaling factors for per-tensor and per-parameter cases) sampled from a log-uniform distribution:

$$\alpha = \exp(u), \quad u \sim \text{Uniform}(\ln \alpha_{\min}, \ln \alpha_{\max}), \tag{10}$$

with the range typically set to $[\alpha_{\min}, \alpha_{\max}] = [0.001, 1000]$. During training, the learned optimizer sees parameters at vastly different scales within each task, which encourages it to learn scale-invariant update rules that generalize better across different initialization schemes and architectures. In our work, we follow the task augmentation approach used in Celo (Moudgil et al., 2025), applying reparameterization at global-level during meta-training to improve generalization to unseen tasks.

**Outer-training loop.** Putting these components together, the complete meta-training procedure for learned optimizers proceeds as follows. At each meta-training iteration, we sample a task $(\mathcal{D}, \mathcal{L}, \boldsymbol{\theta}_0)$ from the task distribution $\mathcal{T}$ and apply task augmentation to obtain reparameterized initial parameters $\tilde{\boldsymbol{\theta}}_0$. We then perform $K$ steps of inner optimization using the current learned optimizer $f_\phi$ on this task, accumulating a truncated meta-objective over these steps. PES is used to compute unbiased gradient estimates $\nabla_\phi J$ for the learned optimizer parameters by evaluating $n$ perturbations $\phi + \boldsymbol{\epsilon}_i$ on the same truncated segment. The learned optimizer parameters are then updated using these meta-gradients (typically with Adam or another outer optimizer). The inner problem state $(\boldsymbol{\theta}_t, \boldsymbol{s}_t)$ is maintained across truncations, and this process repeats for multiple truncations until the total unroll length $T$ is reached or the task is considered complete. This entire process is repeated across many sampled tasks to train the learned optimizer to generalize across the task distribution $\mathcal{T}$. The combination of efficient gradient estimation via PES, task augmentation for improved generalization, and the expressive MLP architecture enables the practical meta-training of learned optimizers that can match or exceed the performance of hand-designed optimizers like Adam on a variety of tasks.

