# OpenReview forum: "Celo2: Towards Learned Optimization Free Lunch"
_ICLR.cc/2026/Conference — ICLR 2026 Poster_

### Official Review · Reviewer_D6de · 2025-10-20

**Soundness:** 3
**Presentation:** 2
**Contribution:** 3
**Rating:** 6
**Confidence:** 2

**Summary:**

The paper proposes a new learned optimizer that improves upon the state-of-the-art VeLO optimizer. It achieves higher performance while requiring less compute and data during training, and it scales effectively to large tasks. The optimizer generates an update direction, after which classical update components (such as orthogonalization, normalization, and step size adjustment) are applied to produce the final parameter update.

**Strengths:**

- The proposed approach simplifies the training process and scales effectively to larger problems.

- The usage and implementation of the optimizer (in Algorithm 1) are well described.

- The paper presents comprehensive comparisons across multiple tasks and datasets of varying sizes, demonstrating consistent improvements.

**Weaknesses:**

- The paper depends on manually tuned step sizes and learning rate schedules. For instance, the authors note that they “test with 7 learning rate values sampled logarithmically between 1e-3 and 1e-5, warmup fraction 0.05 with cosine decay schedule, and plot the best performing hyperparameter setting for each optimizer.” However, only the best-performing configuration is reported, without clarifying the selection criteria. This limits the understanding of tuning sensitivity and comparability—for example how easily the method can be tuned relative to AdamW.

- The training procedure is described briefly and relies on citations rather than detailed explanation. For instance, “We use Persistent Evolutionary Strategies (PES) (Vicol et al., 2021) to meta-train our optimizer with unroll length logarithmically sampled between 100 and 2000 steps.” Re-explaining the main aspects of PES and its role in this work would make the paper more accessible and reproducible.

- It is unclear how specific training hyperparameters are chosen and applied:
  - Which step size is used during meta-training for each unrolled iteration?
  - Is the cosine decay schedule also used during training?
  - Are orthogonalization and normalization applied at training time, or only at evaluation?

These details are essential for reproducibility and for understanding how much the method’s performance depends on them.

**Questions:**

- How sensitive is the proposed optimizer to the choice of step size and learning rate schedule compared to standard optimizers like AdamW? Could you provide results or plots showing performance across the range of tested hyperparameters?

- During meta-training, which step size or learning rate schedule is actually used? Is it fixed, or does it follow the same cosine decay schedule described for evaluation?

- Are the orthogonalization and normalization steps used during the optimizer’s training phase, or only at test time?

---

> ### Author Response · Authors · 2025-11-27
> **Response to Reviewer D6de**
>
> We thank the reviewer for the constructive feedback, we address the main concerns below.
>
> **Clarification on best-performing config, LR sensitivity analysis.** For each optimizer, we plot the best-performing sweep run with respect to final loss and this is a standard practice in learned optimization literature [1,2,3] to avoid clutter in plotting. However, we have added a hyperparameter sensitivity analysis study in Appendix Section A discussing performance obtained with different learning rates and hyperparameter configurations in supervised learning and reinforcement learning setups. As evident from the analysis (Appendix Figure 5,6), Celo2 demonstrates smooth transition in performance from high-performing to low-performing region as learning rate is swept like AdamW.
>
> **Meta-training hyperparameters.** During meta-training, we use AdamW with a constant LR schedule (1e-4) as in prior work in learned optimization [1,2,3]. This is because meta-training isn't the standard supervised learning setup, has higher noise in learning and is closer to RL in which constant learning rate schedule is a common choice. Moreover, it adds another tunable knob in meta-training and thus increases the compute budget required for a thorough study. However, there is scope for further improving the meta-training tuning setup by thoroughly ablating over meta-training optimizers and their hyperparameters, LR schedules, etc to squeeze out even more performance from learned optimizers which we leave for future work. Regarding step-size during meta-training unrolls, we follow standard practice as in prior work [1,2,3,4] of keeping initial step-size in unrolls during meta-training low (0.001) to keep meta-training stable and learned optimizer parameters get adapted accordingly to do optimization well over meta-iterations. Task augmentation [2,3], used in this work, also changes optimization dynamics during meta-training which essentially forces the optimizer to do well under different augmented reparametrized constraints, thus effectively simulating low to high learning rate regimes during meta-training (see Figure 2 in [2] as a representative example).
>
> **Clarification on orthogonalization and normalization steps.** Orthogonalization / normalization are part of optimizer architecture and hence used during both meta-training and meta-testing. These two techniques can also be used directly at test-time with a learned optimizer that is not meta-trained with them, but we show in ablations that meta-training the learned optimizer with these two techniques, thus removing train-test disparity, is even better (Figure 4).
>
> **Re-explaining PES and its role in this work would make the paper more accessible.** We completely agree and thank the reviewer for the suggestion. To address this, we have added "Additional Background" in Appendix (Section D) which can serve as a gentle introduction to learned optimizers. It covers key aspects such as meta-training and optimizer architecture from relevant prior work that our approach builds upon [1,2,3,4].
>
>
> -----
> [1] Metz, Luke, et al. "Practical tradeoffs between memory, compute, and performance in learned optimizers." CoLLAs (2022).
> [2] Moudgil, Abhinav, et al. "Celo: Training Versatile Learned Optimizers on a Compute Diet." TMLR (2025).
> [3] Metz, Luke, et al. "VeLO: Training versatile learned optimizers by scaling up." arXiv preprint arXiv:2211.09760 (2022).
> [4] Vicol, Paul, Luke Metz, and Jascha Sohl-Dickstein. "Unbiased gradient estimation in unrolled computation graphs with persistent evolution strategies." ICML (2021).

---

> > ### Comment · Reviewer_D6de · 2025-11-27
> > **Thanks for the answer**
> >
> > The authors have addressed my concerns and improved the paper. I believe the method is valuable to the community, and the empirical evidence for its efficiency deserves to be highlighted. Accordingly, I am raising my score.

---

> > > ### Author Response · Authors · 2025-12-02
> > > **Thank you!**
> > >
> > > We appreciate the reviewer for taking the time to review our response, and we are glad to see that the reviewer finds our work valuable and deserving of highlight. Many thanks!

---

### Official Review · Reviewer_9J2f · 2025-10-23

**Soundness:** 2
**Presentation:** 2
**Contribution:** 2
**Rating:** 2
**Confidence:** 3

**Summary:**

This paper proposes a new learnable optimizer Celo2. It is intended to outperforms Celo and other alternatives in terms of generalization, scalability and compute efficiency. This work is built on top of prior Celo optimizer  (Moudgil et al., 2025). This paper introduces the following changes:
a. It retains a user-tunable step size, not touching the learned scheduler. This helps the scalability.
b. It learns a small MLP to serve as a drop-in replacement for the standard Adam update rule. This makes the proposed Celo more computing efficient.
c. The proposed algorithm normalizes the MLP's output, which improves the generalization.

Celo2 shows strong results on ImageNet image classification with ViT. In exp with GPT-2 and GPT-3, it also achieves consistent lower validation loss. On Atari benchmark, it is shown effective in a PPO algorithm to learn an RL policy.

**Strengths:**

The proposed Celo2 does seem to simpler, scalable and generalizable based on the experiments.

There are a wide variety of experiments to back up the claims, e.g. vision, language, and RL.

**Weaknesses:**

The proposed method does not provide sound theoretical guarantee, making it feel ad-hoc. This raises significant concerns about its practical application. In particularly, I would be very concerning on how would a practitioner use the proposed Celo2 in real-world application. For example, it is not clear how to replace the Adam with small MLP? It is unclear if this technique was found to work only on specific, "cherry-picked" problems and whether the MLP architecture must be significantly re-tuned when switching to a new problem domain

The experimental results feel incomplete. While reporting validation loss is informative as a preliminary measure, it is not sufficiently convincing on its own. The paper would be much stronger if it included results from more established, concrete benchmarks (e.g., perplexity or BLEU scores for LLMs, not just training/validation loss).

**Questions:**

1. Could the authors clarify how Celo2 is expected to generalize to even larger models? Large models (LLMs/VLMs) have a massive number of parameters, and their optimizer states (like Adam's moments) consume significant GPU memory. How does the proposed MLP-based approach compare in terms of GPU memory efficiency?

2. What is the architecture of the small MLP used for ImageNet classification versus the one used for the LLM experiments? It is not clear how much this optimizer component needs to be adapted when moving between different problem domains.

3. The paper claims the RMS-normalized learned update leads to better generalization, but the mechanism is unclear. Is there a more detailed ablation study or theoretical justification to support this specific design choice?

**Details Of Ethics Concerns:**

No.

---

> ### Author Response · Authors · 2025-11-27
> **Response to Reviewer 9J2f (1/2)**
>
> We thank the reviewer for the constructive feedback, we address the main concerns below.
>
> **How practitioners would use Celo2 in real-world setting, theoretical guarantees.** From a practical real-use perspective, we would like to emphasize that Celo2 has been tested with highly-optimized codebases [1,2,3] as mentioned in implementation details using modern techniques like fully sharded data parallel (FSDP), splash-attention, fast parallel jax implementations for RL, among others. We have updated the appendix to include our experimental setup details (Appendix B) along with an efficient JAX implementation in Optax (Appendix C). As a result, the practitioner has to simply switch Adam with Celo2 like any other standard optimizer and this ease of use has been our key focus as well by using a simple MLP architecture unlike prior work [4,5] which had complex hierarchical architectures*. Our simple MLP design on the other hand is fully compatible with modern training setups as noted above.
>
> Regarding theoretical guarantees -- the guard mechanism from [6] provides a concrete convergence proof by enforcing a fallback update when the learned optimizer’s step violates simple descent conditions. This guarantee applies independently of the learned update rule, so the same guard can be attached to Celo2 to obtain classical convergence behaviour without modifying our architecture. In addition, recent work [7] develops a PAC-Bayesian trajectory bound together with a KL-based convergence framework that shows learned optimizers converge to critical points with high probability under mild assumptions. Their analysis applies to optimizers with stateful, parametric update rules of the form used in Celo2, which makes their theoretical framework directly compatible with our method. Overall, these works can be leveraged to explore formal guarantees of Celo2 and adapt it accordingly, which is a promising direction for future work. Our focus in this work is on practical performance and scalability.
>
> **Perplexity/BLEU scores for LLMs not just training/validation loss.** We report metrics commonly used in the optimization literature, such as validation loss, from which perplexity can be directly computed by taking its exponential. Our results in Figure 1 focus on studying learned optimizers for upstream pretraining of language models, which is the primary focus of our work. For the ViT ImageNet experiments, we also provide the top-1 accuracy metric, and for the RL experiments, we report episode return, both of which are standard in the literature. However, we agree that studying downstream performance for LLMs with respect to different learned optimizers is definitely an interesting venue for future work along the lines of [11].
>
> **Unclear if Celo2 must be tuned for new domain.** Our core focus on strong meta-generalization performance in Celo2 is to allow plug-and-play use by the practitioner, thus avoiding any re-tuning as the reviewer mentioned. We stress-test our optimizer on unseen tasks under strong settings -- our meta-training set is extremely limited (four 8x8 image classification task with MLP network consisting of 32 hidden units) compared to our large-scale meta-testing tasks (1.3B GPT-3, ViT on ImageNet-256, etc) which makes our setup an ideal testbed for testing meta-generalization. We picked our meta-training tasks directly from prior work [5,8] and kept it fixed, thus allowing us to focus strictly from the recipe and not just on having a better meta-training dataset. Our evaluation task selection was simply driven by interest of ML community and not cherry-picked: we considered the standard tasks from popular domains like NLP, CV, and RL and our evaluation tasks are conducted at a substantially larger scale than those in previous works [8,9] on which we build. Moreover, our focus in this paper is to provide a scalable learned optimizer recipe / approach, not an optimizer artifact that works on all kinds of production-scale workloads as mentioned in the draft. We firmly believe this work is a key stepping stone in making learned optimizers ready for such use cases.

---

> > ### Author Response · Authors · 2025-11-27
> > **Response to Reviewer 9J2f (2/2)**
> >
> > **Celo2 generalizing to even larger models, memory overhead clarification.** We show under strong experimental settings that Celo2 generalizes to standard tasks over 10^6 times larger than its meta-training set size. The key contribution of this paper is demonstrating that learned optimizers can scale to realistic billion-scale tasks under limited compute budget and we believe we have shown that firmly. No prior work in learned optimizers has achieved this feat with our meta-training budget, and they often fail to even stably run on unseen tasks [5,9]. That being said, there is always the question of testing on larger tasks than the ones shown in this paper and 1.3B GPT is where we drew the line given our current compute budget. Learned optimization research is quite compute intensive due to different meta-training and meta-testing stages, and drawing reliable conclusions require exhaustive hyperparameter trial runs, which we did. We believe that we have done our best in terms of pushing the generalization capabilities of our optimizer beyond their meta-training distributions (our main claim) and showing through large-diversity of strong experiments on standard tasks that it indeed works (our core evidence). We invite the academic community to build on the stable learned optimizer baseline proposed in this work and push the frontier even further. Regarding memory overhead for large-scale optimization, Celo2 maintains effective Adafactor features in its state, which scale sublinearly with number of parameters in a tensor. Overall, in terms of parameter, Celo2 has ~5.5x overhead compared to Adam which was 3x overhead but it is still effective cost at runtime [8] due to large-cost incurred by forward and backward pass of the model ([9] dives deep into this). However, there is scope to further improve memory cost by utilizing low-rank features or mixed precision accumulators, which we leave for future work.
> >
> > **How much does Celo2 architecture need adaptation between ImageNet and LLM experiments.** The Celo2 architecture and weights are exactly the same across all experiments. Once Celo2 is meta-trained, we simply plug it into the respective task and test. There is no adaptation in Celo2 weights or changes in its architecture across tasks.
> >
> > **Ablation study on RMS norm.** We have provided an additional study in Appendix Section A thoroughly evaluating 5 different optimizer update normalization choices from meta-generalization perspective. Our proposed RMS normalization consistently performs the best (Table 2). As discussed in our approach section, we apply RMS normalization to the MLP output, producing a fully normalized task-invariant learned update rule (inputs are already RMS-normalized to be invariant to input parameter values [5,8,9]) unlike prior work [5,8,9] which uses direct outputs from MLP. We also ablate over other update normalization choices in Appendix Section A such as RMS norm clipping from Adafactor [10] which provides more background on how it removes large-norm updates to cure instabilities. Empirically, per-step RMS normalization yields the strongest generalization performance in our experiments.
> >
> > _*Previous learned optimizer works [4,5,8] had complicated design with computations happening at tensor-level and global-level -- this restricts scaling to large-scale production infrastructure. For instance, as noted in VeLO [5], VeLO optimizer couldn't be tested properly on large-scale language models (7B at the time) because PAX, an efficient large-scale framework on top of JAX, didn't allow interactions in optimizer at tensor-level, since it used an efficient scan over layers to reduce compile times of large models._
> >
> >
> > ----
> > [1] https://github.com/martin-marek/batch-size
> > [2] https://github.com/kvfrans/jaxtransformer
> > [3] https://github.com/luchris429/purejaxrl
> > [4] Wichrowska, Olga, et al. "Learned optimizers that scale and generalize." ICML (2017).
> > [5] Metz, Luke, et al. "VeLO: Training versatile learned optimizers by scaling up." arXiv preprint arXiv:2211.09760 (2022).
> > [6] Prémont-Schwarz, Isabeau, Jaroslav Vítků, and Jan Feyereisl. "A simple guard for learned optimizers." ICML (2022).
> > [7] Sucker, Michael, and Peter Ochs. "A Generalization Result for Convergence in Learning-to-Optimize." ICML (2025).
> > [8] Moudgil, Abhinav, et al. "Celo: Training Versatile Learned Optimizers on a Compute Diet." TMLR (2025).
> > [9] Metz, Luke, et al. "Practical tradeoffs between memory, compute, and performance in learned optimizers." CoLLAs (2022).
> > [10] Shazeer, Noam, and Mitchell Stern. "Adafactor: Adaptive learning rates with sublinear memory cost." ICML (2018).
> > [11] Pascanu, Razvan, et al. "Optimizers qualitatively alter solutions and we should leverage this." arXiv preprint arXiv:2507.12224 (2025).

---

### Official Review · Reviewer_TU4w · 2025-10-27

**Soundness:** 4
**Presentation:** 3
**Contribution:** 3
**Rating:** 4
**Confidence:** 3

**Summary:**

This paper presents a provocative and potentially groundbreaking finding in learned optimization (LO). It directly challenges the prevailing "scaling hypothesis" (e.g., VeLO), which posits that massive meta-training compute is necessary for generalization. The authors introduce Celo2, a learned optimizer meta-trained on a "toy" distribution (8x8 image classification) for a mere 4.5 GPU hours.

The central, surprising claim is that this "cheap" LO, built on a simple normalized MLP architecture and task augmentation, generalizes six orders of magnitude beyond its training data. It successfully and stably trains billion-scale models (GPT-3 1.3B) and vision transformers (ViT), outperforming not only the strong AdamW baseline but also the exorbitantly expensive VeLO, which famously fails on such large-scale tasks.

**Strengths:**

1. The paper's core finding—that a 4.5 GPU-hour "toy" meta-training can produce an LO that scales to 1.3B parameter models—is a "free lunch" that could fundamentally realign research in this field. It moves LOs from a "computationally impossible" (VeLO) to a "highly practical" domain.

2. The paper rightly centers its comparison against VeLO. The results are stark: VeLO, meta-trained with 4000 TPU-months, is unstable and fails on large tasks, while Celo2, trained for 4.5 hours, is stable and superior. This is the paper's strongest point. It demonstrates robust generalization across the three most critical axes:

Model Scale: From tiny 8x8 MLPs to 1.3B GPT-3 models.

Unroll Length: From short 2k-step unrolls to 10B+ tokens (GPT-3) and 50k steps (ViT).

Task Domain: From 2D Image Classification to 1D Language Modeling and Reinforcement Learning (Atari).

3. aThe proposed LO ("Celo2-base") is a small, 8-hidden-unit MLP (Table 1a). The design is explicitly intended as a 1-line "drop-in replacement" for Adam (Sec 4.1), which is a massive win for practical adoption.

**Weaknesses:**

The paper makes extraordinary claims (4.5 GPU-hour meta-training, 6-orders-of-magnitude generalization) with a simple recipe. Such "too good to be true" results demand exceptional evidence. However, the paper is missing an appendix and supplementary material, providing no code, implementation details, or full hyperparameter lists beyond what is in the main text. This makes it impossible to verify the claims or assess reproducibility, which is paramount for such a shocking result.

**Questions:**

1. The decoupling of the learning rate (Sec 3) is a major simplification. How much of Celo2's superior performance and stability can be attributed to relying on a hand-tuned cosine schedule, which VeLO was explicitly designed to avoid?

2. Figure 4 shows that Celo2's best performance comes from hybridization (adding Orthogonalization and AdamW@1D). Does this suggest that the future of LOs is not "pure" end-to-end learning, but rather learning "plugins" that work with hand-designed rules?

3. Could you please expand on the choice of "average loss" vs. "final loss" as the meta-objective? Does Celo2's stability come from optimizing this easier, myopic objective? What happens if Celo2 is meta-trained to optimize the final loss, as VeLO does?

---

> ### Author Response · Authors · 2025-11-27
> **Response to Reviewer TU4w (1/2)**
>
> We thank the reviewer for the constructive feedback, we address the main concerns below.
>
> **Missing details, reproducibility concerns.** We understand reviewer's concern. We have now updated our draft with appendix sections B and C containing JAX implementation, experimental setup details, and all the hyperparameter configs. As mentioned in the main paper text, our benchmarking is thorough, closely following strongly tuning setups from prior work, and we further tune the baselines in our experiments to ensure that comparisons are fair. Moreover, we implement our learned optimizers in standard Optax syntax (Appendix C) which we believe is also a critical contribution and opens up learned optimizers for the academic community to further build on them. We have also added 'Additional Background' in Appendix Section D, providing a concise review of the relevant literature in this area.
>
> **Role of decoupling learning rate in Celo2.** As mentioned in our approach section, decoupling learning rate from optimizer architecture and keeping it tunable as per the task is one of the critical parts of our recipe which allows learned optimizers to generalize from limited compute budget. However, as we show through a bunch of ablations (Table 1), the meta-generalization of a learned optimizer is also heavily dependent on its architecture and meta-training procedure which affects its competitiveness relative to hand-designed optimizers. Before our work, learned optimizers on moderate compute budget (unlike VeLO) were mostly tested on small toy tasks such as Fashion MNIST and CIFAR-10 [1,2,3]. This work makes a major leap in pushing the generalization frontier of learned optimizers by demonstrating that they can also work on billion-scale models by exactly pinpointing the recipe in terms of problem setup (which params should be tunable and meta-trained), meta-training and optimizer architecture choices, hence treating meta-generalization as the first class citizen.
>
> **Comment on LOs future direction (hybridization).** We show that that advances in optimization such as orthogonalization, decoupled weight decay, and others are compatible with learned optimizers and improve their performance just like hand-designed optimizers. However, there is a lot of scope in simply improving the meta-training recipes by training learned optimizers without these techniques on diverse distribution of tasks which can push their performance (literature in learned optimizers has shown this to some extent [2,4,5]). In this work, we keep our meta-training set, fixed and simple, following prior work [2, 5], which allows us to systematically perform large-number of optimizer architecture and meta-training ablations in order to improve them. As a result, we demonstrate that learned optimizers can strongly meta-generalize to large-scale out-of-distribution tasks even on an extremely limited meta-training compute budget (much higher compute efficiency compared to VeLO). We leave systematic study with respect to meta-training dataset / scaling laws for future work, which is within scope now since the compute efficiency of learned optimizers is established in this work already.

---

> > ### Author Response · Authors · 2025-11-27
> > **Response to Reviewer TU4w (2/2)**
> >
> > **Celo2 meta-training average loss vs final loss clarification.** Given the problem setup we propose in which the goal is to learn update rule (Approach section) instead of learning rate schedules, conceptually average loss as meta-objective is more suitable since if we want to learn an update rule that performs the best in every stage of training and allows for flexible usage by the practitioner (e.g. resuming training with any schedule, any training schedules). However, it is not the reason for stability since prior work also trains learned optimizers with the same PES average loss objective but still struggles to meta-generalize due to flaws in meta-training and optimizer architecture which we fix in this work. VeLO [5] has a different goal of achieving lowest final loss within the input target step which they tackle specifically with ES but this "API" of input target step also makes it quite restrictive and unsuitable for fine-tuning / RL / novel setups (also discussed in the original paper [5]). Moreover, like we mention in implementation details section, meta-training with final loss objective is extremely computationally expensive. To give a rough idea, in our current setup each meta-update is done with 50 iterations (truncated update) but ES final loss meta-iteration involves unrolling the optimizer for full unroll length which in our case is 2000 iterations. This makes each meta-update 40x more expensive and since we meta-train for 100K iterations, doing a fair head-to-head comparison of ES final loss vs PES average loss is practically infeasible with our compute budget in this rebuttal.
> >
> >
> > -----
> > [1] Metz, Luke, et al. "Practical tradeoffs between memory, compute, and performance in learned optimizers." CoLLAs (2022).
> > [2] Moudgil, Abhinav, et al. "Celo: Training Versatile Learned Optimizers on a Compute Diet." TMLR (2025).
> > [3] Harrison, James, Luke Metz, and Jascha Sohl-Dickstein. "A closer look at learned optimization: Stability, robustness, and inductive biases." NeurIPS (2022).
> > [4] Metz, Luke, et al. "Tasks, stability, architecture, and compute: Training more effective learned optimizers, and using them to train themselves." arXiv preprint arXiv:2009.11243 (2020).
> > [5] Metz, Luke, et al. "VeLO: Training versatile learned optimizers by scaling up." arXiv preprint arXiv:2211.09760 (2022).

---

> > > ### Comment · Reviewer_TU4w · 2025-11-27
> > >
> > > I carefully read the author's response and the appendix. These updates have indeed alleviated my previous doubts of "too good to be true", so I will raise my score.
> > >
> > > Personally, I think that Celo2 is a "paradigm shifting" work. The author provided explanations for some of the code in the appendix, but I'm even more looking forward to seeing its actual "operational" results. By then, it will truly have the potential to end the decade-long "dominance" of Adam. I will continue to follow its subsequent open-sourcing.

---

> > > > ### Author Response · Authors · 2025-12-02
> > > > **Thank you!**
> > > >
> > > > We appreciate the reviewer for taking an in-depth look and interest in our work. We are delighted to see that the reviewer finds their concerns addressed and shares our enthusiasm for this work and its follow-up -- thank you for all the encouraging and kind words!

---

### Official Review · Reviewer_mn2u · 2025-10-28

**Soundness:** 3
**Presentation:** 2
**Contribution:** 4
**Rating:** 6
**Confidence:** 4

**Summary:**

This paper addresses a major limitation of existing learned optimizers (LOs), which is that they do not meta-generalize well out of distribution, particularly when the optimizee is large or the optimization is long. To mitigate this issue, the authors propose Celo2 with a simple yet effective LO architecture that adopts orthogonalization and RMS normalization, which has been shown to be successful in hand-crafted optimizers. The experiments show that Celo2 meta-trained on relatively small datasets and optimizees in a supervised learning paradigm outperforms strong baselines in three out-of-distribution settings: larger optimizee, longer optimization horizon, and application to reinforcement learning. Detailed ablations are provided to analyze each component of Celo2.

**Strengths:**

The method proposed is simple yet effective, indicating a great potential for generalization and further improvement.

The experimental results provide strong support for the claims.

The contribution of this paper is significant towards the practical application of LOs.

**Weaknesses:**

I suggest avoiding the term "free lunch" in the paper, as the no-free-lunch theorem indicates that the inductive bias found by Celo2 will likely downgrade performance on some tasks, although these tasks may be unlikely in realistic applications.

The paper states, "We would like to emphasize that our primary objective while developing this simple approach for learned optimization was stability." However, no sensitivity analysis is provided.

Algorithm 1 is not referred to in the text.

**Questions:**

The ablations are conducted from a generalization-first perspective in what sense? How does the ablation analysis focus on generalization?

What is the purpose of using both TPUs and GPUs in the experiments?

What is the optimizer used for meta-training?

How many steps of optimization were used for the results shown in Fig. 1?

Why do Celo2-base and Adam have identical wall clock time? Does the MLP application on line 5 of algorithm 1 take significantly higher computation?

What are the batch sizes used in the experiments?

---

> ### Author Response · Authors · 2025-11-27
> **Response to Reviewer mn2u**
>
> We thank the reviewer for the constructive feedback, we address the main concerns below.
>
> **Avoid using "free lunch" in title.** We thank the reviewer for the suggestion and will update it in the camera-ready version. We kept "towards" in the title specifically to avoid over-claiming and to cover the cases the reviewer mentioned while still indicating significant progress in the learned optimization area through this work. However, as mentioned, we will change the title accordingly.
>
> **Primary focus is stability but no sensitivity analysis is provided.** We have added an analysis study in Appendix Section A discussing stability and hyperparameter sensitivity of Celo2 in supervised learning and RL setups. Our experiments show smooth sensitivity with respect to tunable parameter in our optimizer (learning rate) just like standard hand-designed optimizers such as Adam. To give context on stability, one of the major issues in learned optimization is stable optimization behaviour on unseen or out-of-distribution tasks [1,2,3,4,5] and prior works failed to stably optimize larger unseen tasks even with exorbitant meta-training compute budget as in VeLO (4000 TPU months) whereas our proposed learned optimizer demonstrates much stronger out-of-distribution performance without any collapse in learning and transfers performance smoothly from small to large-scale unseen tasks (Figure 1).
>
> **How ablations focuses on generalization.** We conduct all the ablation experiments by taking into account the out-of-distribution performance. Specifically, our meta-training set consists of simple image classification MLP tasks (please refer to Section 4, Appendix Section B more details) but all our ablations test the learned optimizer on language modeling task with 30M GPT architecture with 600M tokens of FineWeb-edu dataset which is out-of-distribution on all axes (dataset, model, loss). Throughout our work, we treat meta-generalization as the principal axis for improvement.
>
> **Purpose of using both TPUs and GPUs in the experiments.** The choice of GPU and TPU was simply based on the compute available to us. Due to shortage of GPUs for all the large-scale evaluations in this work, we switched to using TPUs which we fortunately had access to. It is worth noting that our learned optimizer Optax implementation given in Appendix section C is agnostic to the type of compute node (TPU/GPU) which allows us to seamlessly switch between the two.
>
> **Detail clarifications (batch size, optimization steps, etc).** We thank for the feedback and have addressed the concerns in our updated draft. Please find our responses below to specific questions clarifying all the details:
> * _Algorithm 1 not referred to in text:_ We have updated the text to refer to Algorithm 1 and thank the reviewer for flagging this.
> * _What is the optimizer used for meta-training?_ We used AdamW during meta-training following prior work.
> * _How many steps of optimization were used for the results shown in Fig. 1?_ In Fig 1, the training corresponds to 10K optimization steps with batch size 512 and each sample in the batch containing 2048 tokens (sequence/context length).
> * _What are the batch sizes used in the experiments?_ We used batch size 512 in GPT-3, GPT-2 and ViT experiments and batch size 256 in LM-30M experiments. We have added full hyperparameter configs in Appendix Section B.
> * _Why do Celo2-base and Adam have identical wall clock time?_ Does the MLP application on line 5 of algorithm 1 take significantly higher computation? Yes, in theory, MLP application incurs a higher number of flops, but in practical optimization setup with a large-scale model and large batch size, the cost of forward and backward passes heavily dominate the optimization update step. This makes learned optimizers practically usable, as we also observed in our experiments, resulting in comparable wall-clock times. We refer the reviewer to [2, 3] for further discussion on this topic.
>
> We have updated the draft clarifying all the details above and again, thank the reviewer for their service.
>
>
> -----
> [1] Harrison, James, Luke Metz, and Jascha Sohl-Dickstein. "A closer look at learned optimization: Stability, robustness, and inductive biases." NeurIPS (2022).
> [2] Metz, Luke, et al. "Practical tradeoffs between memory, compute, and performance in learned optimizers." CoLLAs (2022).
> [3] Moudgil, Abhinav, et al. "Celo: Training Versatile Learned Optimizers on a Compute Diet." TMLR (2025).
> [4] Metz, Luke, et al. "VeLO: Training versatile learned optimizers by scaling up." arXiv preprint arXiv:2211.09760 (2022).
> [5] Prémont-Schwarz, Isabeau, Jaroslav Vítků, and Jan Feyereisl. "A simple guard for learned optimizers." ICML (2022).

---

> > ### Comment · Reviewer_mn2u · 2025-11-27
> >
> > Thanks for the authors' response. My concerns are well addressed. I raise my score accordingly.

---

> > > ### Author Response · Authors · 2025-12-02
> > > **Thank you!**
> > >
> > > We appreciate the reviewer for taking the time to review our response and are glad to see that their concerns have been well addressed. Many thanks!

---

### Author Response · Authors · 2025-11-27
**Summary response**

We thank the reviewers for their constructive feedback. We have uploaded an updated draft with the requested additional details and experiments. Following is the summary of our changes:

* **[Reviewer TU4w, D6de, mn2u]** We have clarified all the details about experimental setup in Appendix sections B and C including full JAX implementation, and reproducible hyperparameter configs for all the experiments in the main paper.
* **[Reviewer TU4w, D6de, 9J2f]** As suggested, we added an 'Additional Background' section in Appendix section D, covering relevant prior work, thus giving a clear overview of learned optimizers to help readers unfamiliar with the field better understand our work.
* **[Reviewer 9J2f]** We have added experiments in Appendix section A ablating over different output normalization choices in our optimizer. Results indicate that our proposed RMS normalization variant performs the best from generalization perspective.
* **[Reviewer D6de, mn2u]** We have added an analysis study in Appendix section A demonstrating robustness of Celo2 and its stable optimization behaviour across different hyperparameter configs in both supervised learning and reinforcement learning setups.

Given our responses, additional clarifications and further experiments, we hope that the reviewers would consider increasing their scores since we believe this work makes a sizeable contribution to the field of large-scale learned optimization.

---

### Comment · Area_Chair_A194 · 2025-11-27

Dear reviewers,

The authors have provided detailed responses to your reviews. I would appreciate if you could let both me and the authors know how these responses impact your assessment of the paper.

Best,

AC

---

### Author Response · Authors · 2025-12-02
**Post rebuttal discussion summary**

Dear AC

Due to unfortunate incident this year concerning OpenReview, we're taking this opportunity to summarize the discussion with reviewers in short and present a few highlights.

After our rebuttal to initial reviews, 3/4 reviewers [D6de, mn2u, TU4w] with initial scores 6,6,4 respectively, followed up on our responses. All of them found their concerns well addressed and raised their scores. Following are noteworthy comments by our reviewers after the rebuttal:

**[Reviewer TU4w]**
_"Personally, I think that Celo2 is a "paradigm shifting" work. The author provided explanations for some of the code in the appendix, but I'm even more looking forward to seeing its actual "operational" results. By then, it will truly have the potential to end the decade-long "dominance" of Adam. I will continue to follow its subsequent open-sourcing."_

**[Reviewer D6de]**
_"The authors have addressed my concerns and improved the paper. I believe the method is valuable to the community, and the empirical evidence for its efficiency deserves to be highlighted. Accordingly, I am raising my score."_

**[Reviewer mn2u]**
_"My concerns are well addressed. I raise my score accordingly."_

We hope that you take above context into account in your overall assessment of this work. Thanks for your service.

---

### Meta-Review · Area_Chair_oJzy · 2026-01-08

**Summary:**

The paper introduces a new method for meta learning optimizers, showing remarkable generalization from the meta learning to tasks, to much larger scale problems in other domains. The reviewers are very impressed by the performance of the method (almost "too good to be true"), and the initial reviews raised a number of questions about the details of the method and experiments. However, particularly after the rebuttal, reviewers feel that the paper backs up its claims with convincing experiments.

Given a long history of new optimizers that have failed to unseat Adam, the major outstanding question is what the "real world" performance will be, but the paper deserves to be highlighted at ICLR to encourage the community to explore this.

**Reviewer Concerns:**

The authors have done a good job addressing reviewer concerns.

Reviewers D6de, mn2u and TU4w all state that their concerns have been fully addressed.

For Reviewer 9J2f's concerns:
- Lack of theoretical guarantees
I agree that the theoretical basis for the method is limited, but in practice this could be said for much of the field more generally, and practical performance is normally more significant.

- must be significantly re-tuned when switching to a new problem domain
The authors claim a remarkable degree of generalization from meta training to different problem domains

- reliance on validation loss
Validation loss is standard and the least noisy way to evaluation LLMs

- Scalabiltiy to larger models
The authors argue that the method scales sub linearly in the parameter count

- Ablations for RMS norm
The authors have added these

**Reviewer Scores:**

mn2u: 8 (My concerns are well addressed. I raise my score accordingly.)
TU4w: 8 (they describe it as "paradigm shifting" work)
9J2f: 4  (given the level of concern in the original review, the author response may not have been enough to convince them)
D6de: 8 ("I believe the method is valuable to the community, and the empirical evidence for its efficiency deserves to be highlighted.")

---

### Decision · Program_Chairs · 2026-01-26

Accept (Poster)